# A Meta-learner for Heterogeneous Effects in Difference-in-Differences

Hui Lan [1]   Haoge Chang [2]   Eleanor Dillon [3]   Vasilis Syrgkanis [4]

## Abstract

We address the problem of estimating heterogeneous treatment effects in panel data, adopting the popular Difference-in-Differences (DiD) framework under the conditional parallel trends assumption. We propose a novel doubly robust meta-learner for the Conditional Average Treatment Effect on the Treated (CATT), reducing the estimation to a convex risk minimization problem involving a set of auxiliary models. Our framework allows for the flexible estimation of the CATT, when conditioning on any subset of variables of interest using generic machine learning. Leveraging Neyman orthogonality, our proposed approach is robust to estimation errors in the auxiliary models. As a generalization to our main result, we develop a meta-learning approach for the estimation of general conditional functionals under covariate shift. We also provide an extension to the instrumented DiD setting with non-compliance. Empirical results demonstrate the superiority of our approach over existing baselines.

## 1. Introduction

Difference-in-Differences estimators have become a foundational tool for causal inference in economics (Roth et al., 2023), social sciences (Chiu et al., 2023) and healthcare (Wang et al., 2024) for evaluating causal effects of policy interventions or treatments when both pre- and post-treatment outcomes are observed. In contrast to cross-sectional data, having panel data enables researchers to work with different assumptions that are often considered more plausible in application. Due to the non-random assignment of treatments,

estimating the causal effect of a treatment or intervention in observational studies often requires strong assumptions, such as conditional exogeneity, which rules out unobserved confounding. Panel data consist of repeated observations of the same units over time, which allows researchers to control for certain types of unobserved, time-invariant characteristics. Due to its flexibility and robustness in handling non-experimental data, the DiD approach has gained significant traction in empirical research, especially in the evaluation of policy interventions (e.g. Thome et al., 2024, etc.), labor market changes (e.g. Card & Krueger, 1994; Rossin-Slater et al., 2013; Pierce & Schott, 2016, etc.), environmental regulations (e.g. Gao et al., 2020, etc.), and public health (e.g. Finkelstein et al., 2012; Dimick & Ryan, 2014, etc.).

Despite several recent methodological advances in the DiD literature (Roth et al., 2023; Chiu et al., 2023), most state-of-the-art approaches are still only able to generate average causal effects, or at best group average causal effects for predefined subpopulations. On the contrary, in many empirical applications, especially on large-scale datasets that stem from digital platforms, practitioners are interested in treatment effect heterogeneity for personalized decision making. The estimation of heterogeneous treatment effects has gained considerable attention in recent years due to its potential to uncover variation in how different subpopulations respond to an intervention. Motivated by the success of machine learning techniques in learning complex tasks, many studies have employed them in learning heterogeneous treatment effects, see for instance Shalit et al., 2017; Shi et al., 2019; Künzel et al., 2019; Nie & Wager, 2021; Oprescu et al., 2019; Kennedy, 2023, etc. However, estimating heterogeneous treatment effects for panel data remains relatively unexplored in literature.

In this paper, we explore the estimation of heterogeneous treatment effects of a binary treatment using panel data under the canonical parallel trends condition used in DiD setups (e.g. Ashenfelter & Card, 1984; Card & Krueger, 1994, etc.). The parallel trends assumption posits that, in the absence of treatment, the treated and control units would have followed similar trends over time. Recent research has explored different approaches in addressing limitations of traditional methods (e.g. Roth et al., 2023). One line of work focuses on relaxing the unconditional parallel trends assumption by taking into account systematic differences

*Part of this work is done during an internship at Microsoft Research. **Vasilis Syrgkanis and Hui Lan are Supported by NSF Award IIS-2337916. [1]Institute of Computational and Mathematical Engineering, Stanford University, Stanford, USA [2]Department of Economics, Columbia University [3]Microsoft Research, New England [4]Department of Management Science and Engineering, Stanford University, Stanford, USA. Correspondence to: Hui Lan <huilan@stanford.edu>.

*Proceedings of the $42^{nd}$ International Conference on Machine Learning*, Vancouver, Canada. PMLR 267, 2025. Copyright 2025 by the author(s).

in the time trends due to other (observed) characteristics through the conditional parallel trends condition (e.g. Heckman et al., 1997; Sant'Anna & Zhao, 2020, etc.). Another line of research tackles the challenges of estimating average treatment effects under treatment effect heterogeneity over time for multi-period settings (e.g. Sun & Abraham, 2021; Callaway & Sant'Anna, 2021, etc.). This paper synthesizes the insights from these two lines of works, and extends the framework to incorporate heterogeneous treatment effects across any dimension, in a flexible manner.

We propose a doubly robust estimation framework for the conditional average treatment effect on the treated (CATT), and show that the mean squared error (MSE) of the learned model is robust to the estimation error of auxiliary models that need to be estimated. While there are doubly robust estimators proposed for unconditional ATT with panel data (e.g. Sant'Anna & Zhao, 2020; Callaway & Sant'Anna, 2021), there does not exist one for the heterogeneous effect. In contrast to the conditonal average treatment effect (CATE), the asymmetry of the CATT allows our proposed method to avoid estimating a conditional outcome model under treatment, which can be hard to learn given a unbalanced dataset with a small number of treated units. We also draw the connection to the literature on debiasing under covariate shift (Chernozhukov et al., 2023), and provide an extension of our main result to a unifying framework for general conditional functionals, encompassing many widely encountered empirical problems such as conditional prediction powered inference under co-variate shift, heterogeneous long-term effects via surrogates based on historical data and heterogeneous treatment effects tailored to target sub-populations. Moreover, we extend our main result to the case of a binary instrument (or exposure to treatment) with two-sided non-compliance, and provide a doubly robust estimator for the conditional local average treatment effect of the exposed.

Similar to Ogburn et al., 2015, Semenova & Chernozhukov, 2021 and Oprescu et al., 2019, we consider a framework that allows for the conditional parallel trends assumption to condition on a high dimensional set of observed covariates, denoted as $W$. This conditioning strengthens the plausibility of the assumptions and improves the robustness of the resulting estimators. Our focus is on the estimation of the average treatment effect on the treated (ATT) while conditioning on any subset, $X$, of the covariates $W$. Estimating the projection of heterogeneous treatment effects onto a subset of covariates is particularly advantageous for interpretation, when the goal is to uncover heterogeneity with respect to a set of key features that are of most interest. For instance, in medical applications, we might have high-dimensional imaging data that can be used to predict the outcome, while we are only interested in understanding how the treatment effect is modified by other features such as age, bone density, etc. Furthermore, this framework can be

helpful for decision making when trying to leverage the findings to deploy a personalized policy on a larger population for which only a subset of covariates is available.

We demonstrate using synthetic and semi-synthetic experiments that the proposed meta-learner outperforms prior baselines. Finally, we applied our method on a real-world case study on the effects of raising minimum wage on teen employment. Our flexible doubly robust meta-learner automatically identified dimensions and patterns of heterogeneity that had not been highlighted in prior literature. In particular, our method uncovered that the county population plays a significant role on the magnitude of the treatment effect of raising the minimum wage on teen employment and even though this effect can be quite large and negative for small counties, it becomes negligible and close to zero on large counties. We developed an out-of-sample validation pipeline and showcased that the patterns of heterogeneity identified by our methodology are statistically significant.

## 2. Problem Statement

We consider the standard setup in the DiD framework. We observe a balanced panel with $n$ units and $T$ periods. We denote time by $t = 0, ..., T - 1$. The units are assumed to be an i.i.d sample from a superpopulation. For each unit $i$, we observe a time series of outcomes $\{Y_{it}\}_{t=1}^{T}$, a time series of binary treatment status $\{D_{it}\}_{t=1}^{T}$, and time-invariant covariates $W_i$. For simplicity, we restrict our discussion to $T = 2$ periods in this section and Section 3. We discuss extensions to the multi time period setting in Section 5.

We adopt the potential outcomes framework and assume for unit $i$ at time $t$, the outcome is generated as:

$$Y_{i,t} = D_{i,t}Y_{i,t}(1) + (1 - D_{i,t})Y_{i,t}(0)$$

where $Y_{i,t}(d)$ denotes the potential outcome at time $t$ under treatment $d$. For brevity of notation, we may drop the unit subscript $i$. We assume that both the treated and untreated groups are untreated at $t = 0$, and the treated group becomes treated at $t = 1$, while the control group remains untreated. Our target estimand is the conditional average treatment effect on the treated (CATT), conditioning on any subset $X$ of the covariates $W$:

$$\theta_0(X) = \mathbb{E}[Y_1(1) - Y_1(0)|D = 1, X].$$

### 2.1. Assumptions and Identification

Panel data allows us to disentangle unobserved confounding to some degree by leveraging both cross-sectional and time-series variations. In this section, we focus on the conditional parallel trends assumption that is commonly employed in the empirical literature to identify treatment effects for panel data. This assumption posits that the untreated outcome will

evolve in parallel for both the treated and untreated group, for units with the same observed characteristics $W$.

**Assumption 2.1** (Conditional Parallel Trends).

$$\mathbb{E}[Y_1(0) - Y_0(0)|D_1 = 1, W]$$
$$= \mathbb{E}[Y_1(0) - Y_0(0)|D_1 = 0, W]$$

Conditioning on covariates makes the assumption more plausible, as it allows the treatment assignment to depend on any baseline trends that are predictable from the observed covariates. A practical motivation comes from the abundance of pre-treatment outcome data (for time periods before $t = 0$). It could be reasonable to condition on the full outcome history to try to account for cases where the magnitude of the growth (or decline) through time might depend on the base outcome level. For instance, employees with a higher salary usually receive higher pay raises through time.

**Assumption 2.2** (No-anticipation Assumption).

$$\mathbb{E}[Y_0(0) - Y_0(1)|D_1 = 1, W] = 0$$

In practical applications, Assumption 2.1 is imposed with the full set of covariates $W$ for plausibility, as we expect more covariates to able to capture more confounding. However, we might only be interested in the heterogeneity of the treatment effect in a smaller and interpretable subset of the covariates $X \subset W$.

**Proposition 2.3.** *Under Assumptions 2.1 and 2.2, the CATT, $\theta_0(X)$, can be identified as:*

$$\theta_0(X) = \mathbb{E}[Y_1 - Y_0 - g_0(X)|D = 1, X],$$

*where* $g_0(x) := \mathbb{E}[Y_1(0) - Y_0(0)|D = 0, W = w]$.

## 3. DR-Learner for CATT

In the special case when $W = X$, the statistical problem that results from Proposition 2.3 is identical to the estimation of the conditional average treatment effect under conditional ignorability with outcomes $Y_1 - Y_0$ (even though, the resulting statistical model can only be interpreted as a CATT, due to the one-sided nature of the parallel trends assumption). For discussion, see Appendix C.

However, when $X \subset W$, this equivalence no longer holds and prior approaches for CATE estimation under conditional exogeneity is no longer applicable and can lead to biased results even in the limit of infinite samples. For instance, the simplest identification formula for the CATE and its accompanying estimation estimation strategy, the $T$-Learner, would estimate the statistical model:

$$\tau_0(X) = \mathbb{E}[g_1(W) - g_0(W) \mid X],$$

where $g_d(W) = \mathbb{E}[Y_1 - Y_0 \mid D = d, W]$. However, under the conditional parallel trends assumption it is no longer the case that $\mathbb{E}[Y_1 - Y_0 \mid D = 1, W] = \mathbb{E}[Y_1(1) - Y_0(1) \mid W]$, since the parallel trends assumption crucially does not make any restriction that the trends under treatment are conditionally parallel between treated and control units. Therefore $\mathbb{E}[g_1(W) \mid X] \neq \mathbb{E}[Y_1(1) - Y_0(1) \mid X]$, which subsequently implies that $\tau_0(X) \neq \theta_0(X)$. This difference will be more pronounced for datasets where there is a big difference in the covariate distribution between the treated and un-treated groups.

Thus, when $X \subset W$, the statistical problem that we need to solve based on the identification formula in Proposition 2.3 is inherently different than the statistical problem of estimate a CATE. Hence, we need to develop novel meta-learners, specifically for the CATT, that enjoy local robustness properties analogous to the robustness properties of methods that have been developed for the CATE in prior work (Nie & Wager, 2021; Foster & Syrgkanis, 2023; Oprescu et al., 2019; Kennedy, 2023). Our main result will be a doubly-robust meta learner for the CATT.

The simplest plug-in meta-learning approach for the CATT is to construct an estimate $\hat{g}_0$ of the baseline growth model $g_0$ using generic ML techniques (since it corresponds to the regression problem of predicting the difference $Y_1 - Y_0$ from covariate $W$, using samples only from the control population, i.e., $D = 0$) and then estimate a CATT model by learning a second-stage regression model that predicts the label $Y_1 - Y_0 - \hat{g}_0(W)$ from covariates $X$, using samples only from the treated population, i.e., $D = 1$.

It is well-known (Chernozhukov et al., 2018) that using ML estimators in a plug-in manner may cause large estimation bias due to, for example, regularization and model mis-speification. A doubly-robust estimator alleviates this concern as it is less sensitive to errors in the baseline growth model $\hat{g}_0$, and allows for consistent estimation under weaker statistical conditions.

To present our main result, we need to present a set of preliminary definitions and assumptions. To avoid ill-posed extrapolations between the treated and untreated groups, we need the following overlap condition:

**Assumption 3.1** (Sufficient Overlap). For all $W$, there exist $c > 0$ such that $c \leq \mathbb{P}(D = 1|W) \leq 1 - c$.

A key concept related to robustness is that of Neyman orthogonality:

**Definition 3.2** (Conditional Neyman Othogonality). Let $m(Z; \theta, \eta)$ be a moment for the target estimand $\theta(\cdot)$ with nuisance functions $\eta = (\eta_1, \eta_2, \dots)$. Such moment is Neyman orthogonal if the directional derivatives with respect to all nuisance functions $\eta$ is zero when evaluated at the true

nuisances, i.e.

$$\partial_\eta \mathbb{E}[m(Z;\theta_0,\eta)|W]\Big|_{\eta=\eta_0} = 0$$

**Lemma 3.3** (Doubly Robust CATT on Subspace of Covariates). *Under Assumptions 2.1, 2.2 and 3.1, the true CATT $\theta_0$ is a solution to the following conditional moment equation:*

$$\mathbb{E}\left[\left(\frac{D-\pi_0(W)}{(1-\pi_0(W))}\right)(\Delta Y - g_0(W)) - D\theta(X) \,\Big|\, X\right] = 0$$

*where $\Delta Y = Y_1 - Y_0$, $g_0(W) = \mathbb{E}[\Delta Y|D=0,W]$, $\pi_0(W) = \mathbb{P}(D=1|W)$. Moreover, this moment is conditionally Neyman orthogonal with respect to all nuisance functions (i.e. $\pi(W)$ and $g(W)$).*

*Remark* 3.4. Comparing with the DR-learner (Kennedy, 2023; Chernozhukov et al., 2017) for conditional average treatment effect (CATE), we note that, by refocusing on the CATT, our proposed moment condition no longer requires the estimation of the conditional expectation of the outcome $\Delta Y$ for the treated group w.r.t the high dimensional $W$. This can be especially advantageous in practical settings where there are only a small number of treated units in the panel, making the estimation of the conditional expectation of the treated units difficult. Moreover, we show that simply regressing the CATE pseudo-outcome as in the DR-learner for CATE will give a biased estimate when the treated and control groups have very different distributions. For more details, please refer to Appendix C.

The next key insight of our paper is that the Neyman orthogonal moment restriction from Lemma 3.3 can be turned into a loss minimization problem and models that satisfy the conditional moment restrictions can be equivalently viewed as minimizers of a strongly convex loss function. This insight is crucial in order to turn the statistical problem into a statistical learning theory problem and subsequently into meta-learning estimation strategy, which will allow for the use of generic ML methods for the estimation of $\theta_0$.

**Proposition 3.5.** *Consider the incomplete squared loss:*

$$\mathcal{L}(\theta;\pi_0,g_0) = \mathbb{E}\left[D\theta(X)^2 - 2\widehat{Y}\theta(X)\right]$$

*where $\widehat{Y}(\pi_0,g_0) = \left(\frac{D-\pi_0(W)}{1-\pi_0(W)}\right)(\Delta Y - g_0(W))$. Under the same assumptions as in Lemma 3.3, the minimizer of $\mathcal{L}(\theta;\pi_0,g_0)$ over any hypothesis space $\Theta$ is equivalent to the solution to the best-projection problem of the CATT among the treated:*

$$\min_{\theta\in\Theta} \mathbb{E}[(\theta(X)-\theta_0(X))^2 \mid D=1]$$

Note that this is a convex loss function, which suggests computational tractability and fast statistical learning rates

and allows it to be efficiently solved using any standard optimization solver. Another advantage of the loss minimization approach is that the out-of-sample loss can be used as a metric for model selection over different function classes (Lan & Syrgkanis, 2024). Moreover, as we show next in our main estimation theorem, this loss-based estimator enjoys double robustness properties, in that it leads to fast rates for the CATT if the product of the estimation rates for $\hat{\pi}$ and $\hat{g}$ decays fast enough.

In the theorem below, we use $\hat{\theta}$ to denote a generic estimator that achieves small excess risk with respect to the plug-in loss $\mathcal{L}(\theta;\hat{\pi},\hat{g})$, where $\hat{\pi}$, $\hat{g}$ are nuisance estimates, constructed from an auxiliary dataset (sample-splitting). Note that the problem of achieving a small excess risk with respect to a given loss is a standard statistical learning theory problem and hence many ML techniques can be invoked to provide such a guarantee. Hence, our theorem accommodates estimators resulting from a variety of CATT ML estimators, such as empirical risk minimization on the empirical loss, gradient boosted forests or neural networks.

**Theorem 3.6** (CATT Rates). *Let $\hat{\pi}, \hat{g}$ be estimates of the nuisance functions, constructed using an auxiliary dataset. Let $\|\theta\|_{D=1} = \sqrt{\mathbb{E}[\theta(X)^2|D=1]}$ denote the $L_2$ norm over the treated population. Let $\hat{\theta}$ be the result of any estimation process using $n$ samples, satisfying w.p. $1-\delta$*

$$\mathcal{L}(\hat{\theta};\hat{\pi},\hat{g}) - \inf_{\theta\in\Theta}\mathcal{L}(\theta;\hat{\pi},\hat{g}) \leq R_{n,\delta}^2$$

*Suppose Assumptions 2.1, 2.2, and 3.1. If the hypothesis space $\Theta$ is convex or is well specified (i.e. $\theta_0 \in \Theta$), then $\hat{\theta}$ satisfies w.p. $1-\delta$:*

$$\|\hat{\theta}(X) - \theta_*(X)\|_{D=1}^2 \leq$$
$$\frac{4}{\rho}R_{n,\delta}^2 + \beta\,\mathbb{E}\left[\mathbb{E}\left[(\hat{g}(W)-g_0(W))\left(\frac{\pi_0(W)-\hat{\pi}(W)}{1-\hat{\pi}(W)}\right)\Big|X\right]^2\right]$$

*where $\rho = \mathbb{P}(D=1)$ and $\beta = \frac{2}{\rho^2 c^2}$ and*

$$\theta_* \in \arg\min_{\theta\in\Theta}\|\theta(X)-\theta_0(X)\|_{D=1}^2$$

**Lagged Dependent Outcome Alternate Assumption:** In Appendix B, we also provide an extension of our approach under the lagged dependent outcome assumption, which posits that the past outcomes capture sufficient information to disentangle future outcomes and treatment assignment. This assumption is commonly used to model time series data and is also used in estimating treatment effects (e.g. Angrist & Pischke, 2009; Antonelli et al., 2024, etc.).

**DiD with Instruments:** As a further extension, we consider the setting of estimating heterogeneous effects from panel data with a binary instrument $Z$. This has applications in policy evaluation where the exposure to the policy does not perfectly determine treatment receipt due to

non-compliance (Gerber & Green, 2012), and we are only willing to assume parallel trends on the exposure and not the chosen treatment. Thus, policy exposure can be interpreted as the instrument. In Appendix A, we present a meta-learner for this IV-DID setup.

## 4. General Conditional Functionals Under Covariate Shift

In this section, we show that the CATT estimation problem under conditional parallel trends can be viewed as a special case of a much more broad statistical estimation problem which can capture many other empirical problems beyond heterogeneous effects in DiD analysis.

In particular, we consider the following estimation problem. Consider data consisting of $Z$, which contains covariates $W$ drawn from a target distribution $\mathcal{D}_t$. Let $\mathbb{E}_t[\cdot]$ denote the expectation with respect to the distribution $\mathcal{D}_t$. The goal is to estimate a conditional linear functional $\mathbb{E}_t[m(Z; g_0)|X]$ of the regression function $g_0(W) = \mathbb{E}[Y|W]$, where $X$ is a subset of $W$, $m$ is a linear moment functional of $g_0$ and the expectation is taken with respect to the target distribution. On the other hand, labels for the target variable $Y$ of the regression function are available only on data where the covariates are drawn from a different source distribution, i.e., $(Y, W) \sim \mathcal{D}_s$. Let $\mathbb{E}_s[\cdot]$ denote the expectation with respect to $\mathcal{D}_s$. We assume that there is only covariate drift and no concept drift, i.e.

**Assumption 4.1** (No concept drift). $g_0(W) = \mathbb{E}_s[Y|W] = \mathbb{E}_t[Y|W] = \mathbb{E}[Y|W]$.

Let $E$ denote the indicator variable of whether the sample stems from the target distribution environment. We can then rewrite the statistical estimand as:

$$\theta(X) = \mathbb{E}[m(Z; g)|E = 1, X]$$

For instance, in the case of the CATT problem in the DiD setting, the moment is $m(Z; g) = Y_1(1) - Y_0(1) - g(W)$ and the outcome regression $g(W) = \mathbb{E}[Y_1(0) - Y_0(0)|W, D = 0]$ is learned based on the covariate distribution of the untreated units, while the estimand is the conditional functional $\mathbb{E}[m(Z; g)|X, D = 1] = \mathbb{E}[Y_1 - Y_0 - g(W)|X, D = 1]$, which is a conditional expectation taken over the covariate distribution of the treated units, conditioning on a subset $X$ of $W$. Since the label for the regression function $g(W)$ is $Y_1(0) - Y_0(0)$, it is only available for the untreated group.

Debiasing techniques for unconditional functionals under covariate shift were analyzed in the prior work of Chernozhukov et al., 2023. In this paper, we substantially extend their analysis to the case of conditional functionals and provide a doubly robust meta-learning strategy for any such conditional linear functional problem under covariate shift.

We further motivate this setup with several other empirically prevalent examples from the machine learning and causal inference literature.

**Example 4.2** (Conditional prediction powered inference). In settings where prediction is a central task, it is often desirable to leverage predictive models to improve the efficiency and accuracy of statistical inference. Consider some high-dimensional features or covariates $W$, some labels $Y$, and a simulation model $g(W)$ for the predictive task $\mathbb{E}[Y \mid W]$. An example of the prediction powered inference framework of (Angelopoulos et al., 2023), asks to estimate $\mathbb{E}_t[Y]$. However, we might only have labeled data on a smaller or slightly different sub-population $D_s$. In this case, we can use the simulation model and instead target the statistical estimand $\mathbb{E}_t[g(W)]$, using the labeled data only for debiasing the simulation model. Our work extends this setting to allow for the estimation of conditional means with respect to a subset of the covariates $X$, in the target distribution, i.e. $\theta(X) = \mathbb{E}_t[g(W) \mid X]$ and in a setting where the covariate shift density ratio is unknown (prior work considers only the case of a known covariate shift). In many applications, labels might be expensive to obtain and are only available for a small subpopulation which can be a different covariate distribution from the whole population. This setting fits into the framework with $m(Z; g) = g(W)$.

**Example 4.3** (Heterogeneous long-term effects from short-term experiments using historical data). Here we consider settings where we have run a short-term experiment, where a treatment $D$ was randomized over a population of users drawn from $D_t$ and our goal is to estimate the effect of $D$ on a long-term outcome $Y$. However, we want to estimate that effect without the need to wait for the long-term effect to materialize, but solely based on short-term data. A typical technique used in this setting is the surrogate approach, where we assume that the long-term outcome $Y$, is not directly affected by the treatment $D$, but is affected indirectly through some short-term or "surrogate" post-treatment outcomes $S$, i.e. $Y(d) = Y(S(d))$. Under this assumption, it can be shown that the long-term effect can be identified by measuring the effect of the treatment on the predicted long term outcome, based on the surrogates and other potentially pre-treatment covariates $X$. For any set of pre-treatment co-variates $X$, we can identify the CATE as $\theta(X) = \mathbb{E}[Y(1) - Y(0) \mid X] = \mathbb{E}[g(W) \mid D = 1, X] - \mathbb{E}[g(W) \mid D = 0, X]$, where $W = (S, X)$ and $g(W) = \mathbb{E}[Y \mid W]$. The function $g(W)$ can be learned using historical data where we have access to short-term signals $S$, characteristics $X$ and long term outcomes $Y$. However, the historical covariate distribution $D_s$ can potentially be different from the distribution $D_t$. Since the treatment is randomized, we can write the target estimand

*Table 1.* MSE (mean ± standard deviation) Over 100 Simulations. Each row represent a different meta-learner, and columns represent the different nuisance function classes.

| | Linear Regression | Lasso (CV) | Ridge (CV) | Random Forest | Best |
|---|---|---|---|---|---|
| Neural Net (OR) | 0.12 ± 0.02 | 0.12 ± 0.02 | 0.12 ± 0.02 | 0.38 ± 0.18 | 0.12 ± 0.02 |
| Neural Net (DR) | **0.1 ± 0.02** | **0.1 ± 0.03** | **0.1 ± 0.02** | **0.14 ± 0.04** | **0.1 ± 0.02** |
| XGBoost (OR) | 0.09 ± 0.02 | 0.09 ± 0.02 | 0.09 ± 0.02 | 0.31 ± 0.16 | 0.09 ± 0.02 |
| XGBoost (DR) | **0.04 ± 0.01** | **0.04 ± 0.01** | **0.04 ± 0.02** | **0.06 ± 0.03** | **0.04 ± 0.01** |

as:

$$\theta(X) = \mathbb{E}_s\left[g(W)\left(\frac{D}{\pi} - \frac{1-D}{1-\pi}\right)\Big|X\right]$$

where $\pi = \mathbb{P}(D = 1) = \mathbb{P}(D = 1 \mid W)$. This setting falls in the framework with $m(Z; g) = g(X)\left(\frac{D}{\pi} - \frac{1-D}{1-\pi}\right)$.

**Example 4.4** (CATE with covariate shift). Consider the case of estimating the CATE $\tau_0(X)$ under conditional exogeneity. Many times we want to understand the projection of the CATE on a subset of variables $W$ and over some target population $D_t$ over which we will deploy our personalized policy. However, we might want to use a bigger population $D_s$ to train our CATE model, so as to increase accuracy. In this setting, the target statistical estimand can be written as $\mathbb{E}_t[g(1, W) - g(0, W) \mid X]$, where $g(D, W) = \mathbb{E}[Y \mid D, W]$. This lies in the framework with $m(Z; g) = g(1, W) - g(0, W)$.

We provide a debiasing framework for this problem. Before presenting the main results, we state the necessary definitions and assumptions.

**Definition 4.5** (Conditional Riesz Representer). The Conditional Riesz Representer of a continuous linear functional $m(Z; g)$ on $X$, with respect to some function $g(W)$, is the square-integrable random variable $\alpha(X)$ such that:

$$\mathbb{E}_s[m(Z; g)|X] = \mathbb{E}_s[\alpha(W)g(W)|X]$$
$$\forall g(W) \quad s.t. \quad \mathbb{E}[g(W)^2] < \infty$$

**Assumption 4.6** (Sufficient Overlap Under Covariate Shift). For all $W$, there exist $c > 0$ such that $c \leq \mathbb{P}(E = 1|W) \leq 1 - c$.

**Theorem 4.7** (Neyman Orthogonal Moments for General Conditional Funcitonals under Covariate Shift). *Suppose that Assumptions 4.1 and 4.6 hold. Consider a nuisance regression function $g_0(W) = \mathbb{E}[Y|W]$, and target estimand $\theta(X) = \mathbb{E}_t[m(Z; g_0)|X] = \mathbb{E}[m(Z; g_0)|E = 1, X]$, where $m(Z; g)$ is a continuous linear functional of $g$. The true solution $\theta_0(X)$ satisfies the following conditional moment restriction that is Neyman orthogonal with respect to all the*

*nuisance functions $\pi(W)$, $\alpha(W)$ and $g(W)$:*

$$\mathbb{E}\Big[E \cdot (m(Z; g) - \theta(X)) +$$
$$(1 - E) \cdot \frac{\pi(W)}{1 - \pi(W)}\alpha(W)(Y - g(W))\Big|X\Big] = 0$$

*where $\pi(W) = \mathbb{P}(E = 1|W)$ and $\alpha(W)$ is the conditional Riesz representer of $\mathbb{E}_s[m(Z; g) \mid X]$.*

In the CATT application the Riesz Representer is $-1$. In Example 4.2, the Riesz representer is 1. In Example 4.3 the Riesz representer is $\frac{q(W)}{\pi} + \frac{1-q(W)}{1-\pi}$, where $q(W) = \mathbb{P}(D = 1 \mid W, E = 1)$. In Example 4.4 the Riesz representer $\alpha(D, W)$ is $\frac{D}{\mathbb{P}(D=1|E=1,W)} - \frac{1-D}{\mathbb{P}(D=0|E=1,W)}$.

As in Proposition 3.5, this conditional moment can be turned into a convex doubly robust loss minimization problem.

$$\mathcal{L}(\theta; \pi, g) = \mathbb{E}\left[E\theta(X)^2 - 2\widehat{Y}\theta(X)\right]$$

where $\widehat{Y} = Em(Z; g) + \frac{(1-E)\pi(W)}{1-\pi(W)}\alpha(W)(Y - g(W))$. Note that in this loss function the variable $Y$ is always multiplied by $1 - E$ and therefore it respects the constraint that outcomes $Y$ are only available in the source environment. The double robustness property of the loss will make the resulting estimand robust to estimation errors in the nuisance functions. Analogous to Theorem 3.6, we can prove fast statistical learning rates, for the resulting estimator based on this doubly robust loss. It is easy to verify that for the CATT setting this loss coincides with the loss in Section 3.

## 5. Extension to Multi-Period Setting

In the multiple time period setting, we observe the outcomes for each unit for time periods $t = 0, 1, \ldots, T$. Moreover, assume that no unit is treated at period 0. Consider first the case where all treated units are treated at period $G = 1$ and we assume the conditional parallel trends assumption that for all $t \geq 1$, $\mathbb{E}[Y_t(0) - Y_0(0) \mid D = 1, W] = \mathbb{E}[Y_t(0) - Y_0(0) \mid D = 0, W]$. Note that in this case, we can treat the distance $\Delta \in \{0, \ldots, T - 1\}$ of a target period $t$ from the initial treatment time period 1 as a random variable. We can also denote with $Y_{post}(0)$ as the random variable corresponding to the post-treatment period outcome we are

*Table 2.* MSE (mean ± standard deviation) Over 100 Simulations of Imbalanced Dataset. Each row represent a different meta-learner, and columns represent the different nuisance function classes.

| | Linear Regression | Lasso (CV) | Ridge (CV) | Random Forest | Best |
|---|---|---|---|---|---|
| Neural Net (OR) | 0.22 ± 0.06 | 0.21 ± 0.06 | 0.21 ± 0.06 | 0.4 ± 0.15 | 0.21 ± 0.05 |
| Neural Net (DR) | **0.18 ± 0.07** | **0.18 ± 0.05** | **0.18 ± 0.05** | **0.24 ± 0.07** | **0.18 ± 0.05** |
| Neural Net (CATE OR) | 0.27 ± 0.08 | 0.27 ± 0.08 | 0.27 ± 0.08 | 0.51 ± 0.16 | 0.27 ± 0.08 |
| Neural Net (CATE DR) | 0.22 ± 0.07 | 0.22 ± 0.07 | 0.21 ± 0.07 | 0.33 ± 0.11 | 0.21 ± 0.07 |
| XGBoost (OR) | 0.21 ± 0.06 | 0.21 ± 0.06 | 0.21 ± 0.06 | 0.34 ± 0.11 | 0.21 ± 0.06 |
| XGBoost (DR) | **0.12 ± 0.03** | **0.12 ± 0.03** | **0.12 ± 0.03** | **0.18 ± 0.06** | **0.12 ± 0.03** |
| XGBoost (CATE OR) | 0.27 ± 0.08 | 0.27 ± 0.08 | 0.27 ± 0.08 | 0.51 ± 0.16 | 0.27 ± 0.08 |
| XGBoost (CATE DR) | 0.15 ± 0.05 | 0.15 ± 0.04 | 0.15 ± 0.05 | 0.34 ± 0.13 | 0.15 ± 0.04 |

*Table 3.* MSE (mean ± standard deviation) over 100 semi-synthetic datasets generated from the Minimum Wage dataset.

| | Linear Regression | Lasso (CV) | Ridge (CV) | Random Forest | Best |
|---|---|---|---|---|---|
| XGBoost (OR) | 1.97 ± 0.04 | 2.02 ± 0.05 | 1.96 ± 0.04 | 2.08 ± 0.09 | 2.07 ± 0.09 |
| XGBoost (DR) | **1.91 ± 0.04** | **1.88 ± 0.04** | **1.91 ± 0.04** | **1.8 ± 0.09** | **1.8 ± 0.09** |
| XGBoost (CATE OR) | 2.72 ± 0.06 | 2.71 ± 0.07 | 2.73 ± 0.06 | 3.4 ± 0.36 | 2.69 ± 0.07 |
| XGBoost (CATE DR) | 2.73 ± 0.06 | 2.7 ± 0.06 | 2.73 ± 0.07 | 3.47 ± 0.3 | 2.66 ± 0.07 |
| Linear (OR) | 1.96 ± 0.04 | 2.01 ± 0.04 | 1.96 ± 0.04 | 2.07 ± 0.08 | 2.06 ± 0.08 |
| Linear (DR) | **1.92 ± 0.04** | **1.89 ± 0.04** | **1.92 ± 0.04** | **1.83 ± 0.07** | **1.83 ± 0.07** |
| Linear (CATE OR) | 2.78 ± 0.05 | 2.76 ± 0.05 | 2.78 ± 0.05 | 3.14 ± 0.38 | 2.76 ± 0.05 |
| Linear (CATE DR) | 2.8 ± 0.05 | 2.75 ± 0.05 | 2.8 ± 0.05 | 3.13 ± 0.36 | 2.71 ± 0.05 |

looking at. Then we can equivalently write:

$$\mathbb{E}[Y_{post}(1) - Y_{post}(0) \mid X, \Delta = t] = \mathbb{E}[Y_t(1) - Y_t(0) \mid X]$$

Thus we can treat the distance from treatment $\Delta$, as yet another covariate in our framework and make it part of $X$. This way, we can flexibly estimate treatment effect heterogeneity as a function of the distance from the initial treatment period and let ML methods select the best model on how distance from initial treatment changes the effect.

Next consider the more general setting of a staggered roll-out, i.e. each treated unit ($D = 1$) is treated at some period $G = [1, T]$ and remains treated after that period. We denote the never-treated group as $G = \infty$. In this setting, we can make the parallel trends assumption that for all $g \in [1, T]$ and for all $t \geq g$ $\mathbb{E}[Y_t(0) - Y_0(0) \mid D = 1, W, G = g] = \mathbb{E}[Y_t(0) - Y_0(0) \mid D = 0, W, G = \infty]$. Similarly, we can incorporate heterogeneity as a function of the initial treatment period $G$ and the distance $\Delta$ to the treatment period, by making these variables as part of our heterogeneity set $X$: for all $g \in [1, T]$ and $t \in [g, T]$:

$$\mathbb{E}[Y_{post}(1) - Y_{post}(0) \mid D = 1, X, \Delta = \ell, G = g]$$
$$= \mathbb{E}[Y_{g+\ell}(1) - Y_{g+\ell}(0) \mid D = 1, X, G = g]$$

Prior work of Callaway & Sant'Anna, 2021 considers heterogeneity with respect to $G$ and $\Delta$, albeit in a fully non-parametric manner and does not provide a method for model selection, so as to uncover in a more data-driven manner the functional form of this heterogeneity. We note that the prior

work of Callaway & Sant'Anna, 2021 also considers doubly robust estimation and inference on weighted averages of these heterogeneous effect models across different values of $G$ and $\Delta$, which we do not discuss in this work.

## 6. Experiments and Results

### 6.1. Fully Synthetic Data

First, to compare the results of our proposed method with other baselines, we conducted fully simulated experiments where the datasets are generated from known data generating processes that satisfy the identifying assumptions described in Section 2.1. The data has 20 covariates, and the CATT learners look at the projection onto 5 covariates. We report the mean MSE (mean square error) between the predicted CATT and the true CATT on covariates of the treated units of a held out test set. We compare our results with the following baseline models, here $g_d(W) = \mathbb{E}[Y_1 - Y_0 | W, D = d]$, $g(W, D) = g_1(W)D + g_0(W)(1 - D)$, and $\pi(W) = \mathbb{P}(D = 1|W)$:

- Outcome regression (OR) learner: $\theta(X) = \mathbb{E}[Y_1 - Y_0 - g_0(W)|D = 1, X]$

- CATE outcome regression learner: $\theta(X) = \mathbb{E}[g_1(W) - g_0(W)|D = 1, X]$

- CATE DR-learner: $\theta(X) = \mathbb{E}[g(W, 1) - g(W, 0) + \left(\frac{D}{\pi(W)} - \frac{1-D}{1-\pi(W)}\right)(Y - g(W, D))|D = 1, X]$

We considered three different final-stage models for the CATT: neural net, XGBoost, and linear models, to fit the meta-learners. Simulation results are presented in Table 1. The results of linear models can be found in the Appendix 6. The columns represent different ML methods that are used to learn the outcome regression. The propensity function, i.e. $\mathbb{P}(D = 1|W)$, is always fitted using logistic regression. The "Best" column, represents using the ML method that achieved the lowest out-of-sample MSE for the outcome regression. In the Appendix, we also provide results that investigate the performance of our DiD CATT method and the baselines even when the parallel trends assumption is violated. The results in Appendix E suggest that the doubly robust estimator reduces the MSE, as compared to the baselines, even under the violation of parallel trends.

Moreover, we also consider unbalanced datasets, where the size of the control group is much larger than that of the treated group. In particular, the propensities of each unit was lowered by a factor of 10. The results are presented in Table 2. We see that while all models suffered in performance, our proposed doubly robust model still outperforms the other meta-learners as it leverages the asymmetry of the CATT definition to be more robust to unbalanced settings. Notably, the proposed learner out-performs the doubly robust CATE learner as discussed in Remark 3.4.

### 6.2. Minimum Wage Case Study

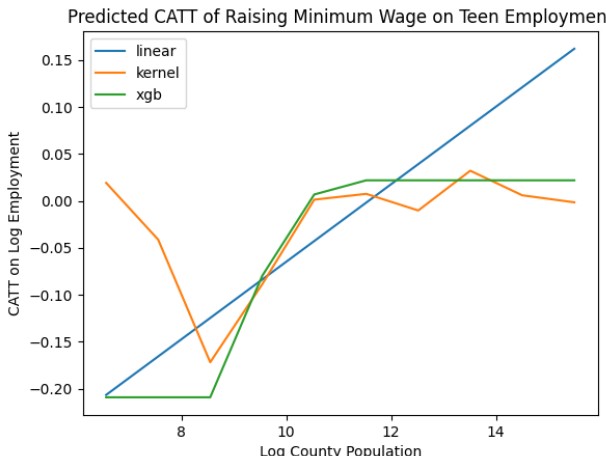

Figure 1. Predicted CATT with respect to log county population.

We applied our proposed approach to the minimum wage dataset that is also studied in Callaway & Sant'Anna, 2021 and Callaway, 2023. This dataset studies the effect of minimum wage changes on teen employment during the period 2001–2007. The outcome variable of interest is the log of county-level teen employment, while the treatment variable is defined as a binary indicator representing whether

a county's minimum wage exceeds the federal minimum wage. The dataset includes covariates such as county population and average annual pay, which serve as controls to account for differences in local economic conditions. For the ease of interpretation, we focus only on the raise in minium wage at year 2004 as the treatment. In our analysis, we treat years after treatment assignment time (2004) as an additional covariate to control for, as discussed in Section 5. Employment rates as well as other covariates from before 2003 are also used as the covariates.

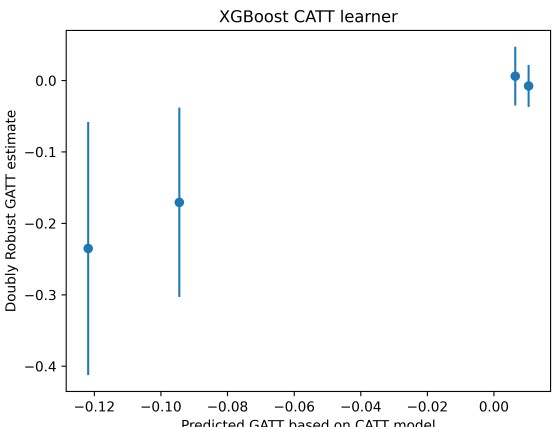

Figure 2. Calibration plot for CATT of minimum wage with respect to log county population.

As a preliminary evaluation, we tested the performance of our methods on semi-synthetic data generated from this dataset. The semi-synthetic data was generated by bootstrapping the samples in the dataset and applying a chosen function for treatment assignment and treatment effect to compute the post-treatment outcomes. Since this dataset is rather low dimensional, we did not experiment with neural networks. The mean MSE with respect to the true CATT function is reported in Table 3. The results show that the proposed method out-performs outcome regression learners as well as the CATE learners.

We then applied our method to the original real dataset. Figure 1 shows the CATT prediction over different values of log county population. The three models used (linear regression, XGBoost, and kernel ridge regression) all showed some extend of positive trends. This suggests that raising minimum wage might have a smaller negative effect on teen employment for counties with a larger population.

**Validating CATT:** Since we do not have access to ground truth treatment effects for real datasets, we need a way to validate the heterogeneity that is picked up by the model is not due to noise. One approach is through calibration. The first step of the procedure is quantile binning the CATT predictions on a held out validation set. Next, the CATT

predictions on a held out test set will be put into the bins according to the tresholds. The group average treatment effect on the treated (GATT) for each bin is calculated as the mean of the heterogeneous model predictions, as well as calculating the unconditional ATT for each group (i.e. conditioning on the empty set, we get $\theta = \mathbb{E}[\widehat{Y}]/\mathbb{E}[D]$). If the heterogeneity is indeed significant, we expect the calibration plots line up in the $45°$ line with non-overlapping confidence intervals.

Figure 2 presents the calibration plot for the doubly robust CATT learner realized using XGBoost. While we see that the GATT for the lowest quantile has a larger confidence intervals, the highest most two quantiles have non-overlapping confidence intervals. This suggests that there is significant heterogeneity between high and low populations. Together with Figure 1, the results seem to suggest that the treatment effect for counties with large populations is close to zero.

## Impact Statement

This paper presents work whose goal is to advance the field of Causal Inference and Machine Learning. There are many potential societal consequences of our work, none which we feel must be specifically highlighted here.

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

# A. DiD with Instruments

As an extension, we consider a widely encountered setting of estimating heterogeneous treatment effects from panel data with a binary instrument $Z$, with two sided non-compliance. In this setting, the target estimand is the conditional local average treatment effect among the exposed (CLATT) in the second (post-treatment) time period:

$$\theta_0(X) = \mathbb{E}[Y_1(1) - Y_1(0) \mid D_1(1) > D_1(0), Z = 1, X]$$

First, we consider the natural conditional extensions, which allows for more heterogeneity and flexibility, of the parallel trends assumptions stated in Miyaji, 2024.

**Assumption A.1** (No carryover assumption). Let $\mathbf{d} = (d_0, d_1)$ denote the treatment path, then $Y_0(\mathbf{d}, z) = Y_0(d_0, z)$ and $Y_1(\mathbf{d}, z) = Y_1(d_1, z)$.

This assumption requires that the outcome is only affected by the current treatment, in other words, there is no carry over effects from previous treatments.

**Assumption A.2** (Exclusion restriction for potential outcomes). For all $t$, $Y_t(\mathbf{d}, z) = Y_t(\mathbf{d})$

This is the standard exclusion restriction assumption for instrumental variables that the instrument only affects the outcome through the treatment.

**Assumption A.3** (Monotonicity Assumption). $\mathbb{P}(D_1(1) \geq D_1(0)) = 1$ or $\mathbb{P}(D_1(1) \leq D_1(0)) = 1$

This assumption requires that the effect of the instrument is monotone - that it either increases treatment adoption or decreases treatment adoption, but not both. This assumption is needed for identification under two-sided non-compliance.

**Assumption A.4** (No anticipation in treatment). $D_0(1) = D_0(0)$ for all units with $Z = 1$

Similar to the standard no-anticipation assumption that the treatment assignment in the second period should not have an affect on the outcome in the first period, here we assume that the exposure event that happens at the second period should not have an anticipatory effect on the treatment adoption in the first period.

**Assumption A.5** (CPTA in Treatment).

$$\mathbb{E}[D_1(0) - D_0(0)|Z = 0, W]$$
$$= \mathbb{E}[D_1(0) - D_0(0)|Z = 1, W]$$

Here we no longer require the instrument to be independent with the potential outcome of the treatments, but instead require that the trend under no exposure is (mean) independent to the exposure.

**Assumption A.6** (CPTA in Outcome).

$$\mathbb{E}[Y_1(D_1(0)) - Y_0(D_0(0))|Z = 0, W]$$
$$= \mathbb{E}[Y_1(D_1(0)) - Y_0(D_0(0))|Z = 1, W]$$

**Assumption A.7** (Sufficient Overlap in Instrument). For all $W$, there exist $c > 0$ such that $c \leq \mathbb{P}(Z = 1|W) \leq 1 - c$.

Moreover if the instrument does not have any effects on the treatment, the local average treatment effect will also not be identified. Hence, we need the following assumption to low-bound the effects of the instrument on the treatment.

**Assumption A.8** (Strong Instrument under PTA). There exist $c_z > 0$ such that

$$\left| \mathbb{E}[D_1 - D_0 - \mathbb{E}[D_1 - D_0 \mid Z = 0, W] \mid Z = 1, X] \right| \geq c_z$$

**Proposition A.9.** *Under Assumptions A.2, A.3, A.4, A.5, A.6, and A.8, the CLATE, $\theta_0(W)$, can be identified as:*

$$\theta_0(W) = \frac{\mathbb{E}[Y_1 - Y_0 - \mathbb{E}[Y_1 - Y_0 \mid Z = 0, W] \mid Z = 1, X]}{\mathbb{E}[D_1 - D_0 - \mathbb{E}[D_1 - D_0 \mid Z = 0, W] \mid Z = 1, X]}$$

In the special case that $W = X$ under the parallel trends assumption, the problem is again equivalent to the standard IV problem, and Syrgkanis et al., 2019 proposed a doubly robust algorithm for estimating the heterogeneous LATE. For more discussion, see Appendix C.

**Lemma A.10** (Doubly Robust Conditional Moment Restriction for CLATE). *Under Assumptions A.2, A.3, A.4, A.5, and A.6, A.7, the true CLATE is a solution to the following conditional moment equation:*

$$\mathbb{E}\left[\widehat{Z}\left\{(\Delta Y - g_Y(W)) - (\Delta Y - g_D(W))\theta(X)\right\} \Big| X\right] = 0$$

*where $\Delta S = S_1 - S_0$ for $S = Y$ or $D$, $g_S(W) = \mathbb{E}[S_1 - S_0 | Z = 0, W]$, and $\widehat{Z} = \frac{Z - \mathbb{P}(Z=1|W)}{1 - \mathbb{P}(Z=1|W)}$. Moreover, this moment is Neyman orthogonal with respect to all nuisance functions.*

**Proposition A.11.** *Consider the incomplete squared loss:*

$$\mathcal{L}_{IV}(\theta; \pi_0, g_{0,Y}, g_{0,D})$$
$$= \mathbb{E}\left[\widehat{Z}\left\{(\Delta D - g_{0,D}(W))\theta(X)^2 - 2(\Delta Y - g_{0,Y}(W))\theta(X)\right\}\right]$$

*where$\Delta S = S_1 - S_0$ for $S = Y$ or $D$, $g_{0,S}(W) = \mathbb{E}[S_1 - S_0 | D = 0, W]$, and $\widehat{Z} = \frac{Z - \pi_0(W)}{1 - \pi_0(W)}$ for $\pi_0(W) = \mathbb{P}(Z = 1|W)$. Under the same assumptions as in Lemma A.10, the minimizer of $\mathcal{L}(\theta; \pi_0, g_{0,Y}, g_{0,D})$ over any hypothesis space $\Theta$ is equivalent to the solution to the best-projection problem of the CATT among the treated:*

$$\min_{\theta \in \Theta} \mathbb{E}[(\theta(X) - \theta_0(X))^2 \mid Z = 1, D(1) > D(0)]$$

**Theorem A.12** (CLATE Rates). *Let $\hat{\pi}, \hat{g}_D, \hat{g}_Y$ be estimates of the nuisance functions, constructed using an auxiliary dataset. Let $\|\theta\|_{D=1,CM} = \sqrt{\mathbb{E}[\theta(X)^2 | D = 1, D(1) > D(0)]}$ denote the $L_2$ norm over the compliers among the treated population. Let $\hat{\theta}$ be the result of any estimation process using $n$ samples, satisfying w.p. $1 - \delta$:*

$$\mathbb{E}\left[\mathcal{L}_{IV}(\hat{\theta}; \hat{\eta}) - \inf_{\theta \in \Theta} \mathcal{L}_{IV}(\theta; \hat{\eta})\right] \leq R_{n,\delta}^2$$

*Define the nuisance errors to be:*

$$Error(\pi, g_D) := \mathbb{E}\left[\mathbb{E}\left[\left(\frac{\pi_0(W) - \pi(W)}{1 - \pi(W)}\right)(g_{0,D}(W) - g_D(W)) \Big| X\right]^2\right]^{\frac{1}{2}}$$

$$Error(\pi, g_Y) := \mathbb{E}\left[\mathbb{E}\left[\left(\frac{\pi_0(W) - \pi(W)}{1 - \pi(W)}\right)(g_{0,Y}(W) - g_Y(W)) \Big| X\right]^2\right]^{\frac{1}{2}}$$

*Suppose Assumptions A.2, A.3, A.4, A.5, and A.6, A.7 are satisfied. Moreover, assume that there exist finite constant $B$ such that $|\theta(X)| \leq B$ for all $X$ with positive measure and all $\theta \in \Theta$. If the hypothesis space $\Theta$ is convex or is well specified (i.e. $\theta_0 \in \Theta$), and $Error(\pi, g_D)$ is sufficiently small ($Error(\pi, g_D) \leq \frac{chk}{8B^2}$), then $\theta$ satisfies, w.p. $1 - \delta$:*

$$\|\theta(X) - \theta_*(X)\|_{D=1,CM}^2 \leq \frac{4}{hk - \frac{8B^2}{c}Error(\pi, g_D)}R_n^2 + \left(\frac{\max(4B^2, 2)}{c\left(hk - \frac{8B^2}{c}Error(\pi, g_D)\right)}\right)(Error(\pi, \hat{g}_Y) + Error(\pi, \hat{g}_D))$$

*where $h = \mathbb{P}(Z = 1)$, $k = \mathbb{P}(D(1) > D(0)|Z = 1)$, and*

$$\theta_* \in \arg\min_{\theta \in \Theta} \|\theta(X) - \theta_0(X)\|_\Theta^2$$

## B. Alternate Assumptions: Lagged Dependent Outcome

The parallel trends assumption guards against linear, time invariant, additive confounding. However, this may be unrealistic in practice. For instance, it might be sensible for the increment with respect to time to depend on the initial level of the outcome, i.e. $Y_1(0) - Y_0(0) \propto Y_0(0)$. One may also be interested in the natural extension to the parallel trends assumptions that also accounts for more complicating confounding pattern. An popular alternative to model panel data is through the lagged dependent variable assumption.

**Definition B.1** (Outcome Support). Let $\mathbb{Y}_{dt}$ denote the support of the outcome at time $t$ for the cohort with treatment assignment $d$.

**Assumption B.2** (Lagged Dependent Outcome with Covariates). $\mathbb{E}[Y_1(0)|Y_0(0) = y, D = 1, W] = \mathbb{E}[Y_1(0)|Y_0(0) = y, D = 0, W]$ for all $y \in \mathbb{Y}_{00}$, and $x \in \mathbb{X}$.

Note that this assumption may be seen as a special case of Assumption 2.1, where the conditioning variable also includes the pre-treatment outcome as well as the observed covariates. This assumption might be more convincing in some practical applications. For instance, in wage studies, current income is generally believed to he highly dependent on past income. However, in cases where the distribution of outcome is significantly different between the treated and untreated groups, this assumption might lead an increase in bias due to matching the pre-treatment outcomes, as shown in (Daw & Hatfield, 2018).

**Assumption B.3** (Overlap in Pre-treatment Outcome). $\mathbb{Y}_{10} \subseteq \mathbb{Y}_{00}$

Note this is a testable assumption, and one can also perform a diagnostic test to ensure that the treated and control outcome distributions have sufficient overlap.

**Proposition B.4.** *Under Assumptions B.2 and B.3, the CATT, $\theta_0(W)$, can be identified as:*

$$\theta_0(X) = \mathbb{E}[Y_1(1) - Y_1(0)|D = 1, X] = \mathbb{E}[Y_1|D = 1, X] - \mathbb{E}[g(Y_0, W)|D = 1, X]$$

*where $g(y, W) = \mathbb{E}[Y_1|D = 0, Y_0 = y, W]$*

*Proof.*

$$\begin{aligned}
\mathbb{E}[Y_1(0)|D = 1, X] &= \int_{\mathbb{Y}_{10}} \mathbb{E}[Y_1(0)|Y_0 = y, D = 1, W]p(Y_0 = y, W|D = 1, X)dy \\
&= \int_{\mathbb{Y}_{10}} \mathbb{E}[Y_1(0)|Y_0 = y, D = 0, W]p(Y_0 = y, W|D = 1, X)dy \\
&= \int_{\mathbb{Y}_{10}} \mathbb{E}[Y_1|Y_0 = y, D = 0, W]p(Y_0 = y, W|D = 1, X)dy \\
&= \mathbb{E}[g(Y_0, X)|D = 1, X]
\end{aligned}$$

$\square$

Similarly, we may also replace the parallel assumptions for IV-CATT by the conditional lagged dependent variable assumptions for both the outcome and treatment.

**Assumption B.5** (Lagged Treatment).

$$\mathbb{E}[D_1(0)|Z = 0, W, D_0] = \mathbb{E}[D_1(0)|Z = 1, W, D_0]$$

**Assumption B.6** (Lagged Outcome).

$$\mathbb{E}[Y_1(D_1(0))|Z = 0, W, Y_0] = \mathbb{E}[Y_1(D_1(0))|Z = 1, W, Y_0]$$

Under the lagged outcome framework, we need a slightly different notion of strong instruments as in the PTA framework.

**Assumption B.7** (Strong Instrument under Lagged Outcome). There exist $c_z > 0$ such that

$$|\mathbb{E}[D_1 - \mathbb{E}[D_1|Z = 1, W, D_0]|Z = 1, X]| \geq c_z$$

**Proposition B.8.** *Under Assumptions A.2, A.3, A.4, B.5, and B.6, the CLATE, $\theta_0(W)$, can be identified as:*

$$\theta_0(X) = \frac{\mathbb{E}[Y_1 - \mathbb{E}[Y_1 \mid Z = 0, Y_0, W] \mid Z = 1, X]}{\mathbb{E}[D_1 - \mathbb{E}[D_1 \mid Z = 0, D_0, W] \mid Z = 1, X]}$$

**Unifying the two assumptions:** First we observe that both Proposition 2.3 and B.4 shares the same general form:

$$\theta_0(X) = \mathbb{E}[S - g(V)|D = 1, X]$$

where $S$ is an observed outcome random variable and $g(V)$ is a nuisance function that is the conditional expectation $\mathbb{E}[S|D = 0, V]$ on some covariates $V$, which is a superset of $X$. Under the parallel trends assumption, $S = Y_1 - Y_0$ and $V = W$, and under the lagged outcome assumption, $S = Y_1$ and $V = [W, Y_0]$. Thus, we see that the results in Section 3 can be generalized to the lagged dependent variable assumption. For IV-DID, similarly to the standard DiD case, when considering $X \subset W$, we can rewrite the identification in Proposition A.9 so that the CLATE is the solution as:

$$\theta_0(X) = \frac{\mathbb{E}[S_Y - g_Y(V_Y) \mid Z = 1, X]}{\mathbb{E}[S_D - g_D(V_D) \mid Z = 1, X]}$$

where $S_r$ is an observed outcome random variable for $r = Y, D$, and $g_r(V)$ is a nuisance function that is the conditional expectation $\mathbb{E}[S_r|Z = 0, V]$ on some covariates $V$, which is a superset of $X$.

## C. Conditioning on the full set of $W$

### C.1. Standard DID

Here, we consider the case where $X = W$, and are interested in estimating:

$$\theta_0(W) = \mathbb{E}[Y_1(1) - Y_1(0)|D_1 = 1, W]$$

Leveraging the no-anticipation and (conditional) parallel trends assumption, we can identify this as:

$$\theta_0(W) = \mathbb{E}[Y_1 - Y_0|D_1 = 1, W] - \mathbb{E}[Y_1 - Y_0|D_1 = 0, W]$$

Note that this shares the same form of identification with the conditional average treatment effect (CATE) under conditional ignorability, but with the differences as the outcome:

$$CATE = \mathbb{E}[Y(1) - Y(0)|W] = \mathbb{E}[Y|W, D = 1] - \mathbb{E}[Y|W, D = 0]$$

In other words, the meta-learners of CATT can be constructed the same way that was constructed for CATE using the difference in outcome. For instance, the doubly-robust pseudo-outcome can be constructed as:

$$Y^{DR} = g(1, W) - g(0, W) + \left(\frac{D}{\pi(W)} - \frac{1 - D}{1 - \pi(W)}\right)(Y_1 - Y_0 - g(D, W))$$

where $g(D, W)$ is an estimator for the conditional expectation $\mathbb{E}[Y_1 - Y_0|W, D]$, and $\pi(W)$ is an estimator of the propensity $\mathbb{P}(D = 1|W)$. The nuisance functions $g(D, W)$ and $\pi(W)$ may be estimated using any ML methods.

As in (Kennedy, 2023), the doubly robust learner (DR-learner) can be constructed as $\theta(W) = \mathbb{E}[Y^{DR}|W]$. Note that due to the asymmetry of the parallel trends assumption, this estimator gives the conditional average treatment effect of the treated. If we further assume that the treatment effects are also mean independent of the treatment conditional on $W$ (i.e. $\mathbb{E}[Y_1(1) - Y_1(0)|D = 1, W] = \mathbb{E}[Y_1(1) - Y_1(0)|D = 0, W]$), then the CATE estimator on the difference in the outcomes identifies the conditional average treatment effects.

However, this CATE pseudo-outcome will give the biased estimate of the CATT when projecting on a subset of covariates, i.e. $\theta^{CATE}(X) = \mathbb{E}[Y^{DR}|X]$ where $X \subset W$. Here we see that $\theta^{CATE}(X) = \mathbb{E}[\mathbb{E}[Y^{DR}|W]|X] = \mathbb{E}[\mathbb{E}[Y_1(1) - Y_1(0)|D = 1, W]|X] \neq \mathbb{E}[\mathbb{E}[Y_1(1) - Y_1(0)|D = 1, W]|D = 1, X] = \mathbb{E}[Y_1(1) - Y_1(0)|D_1 = 1, X]$. This difference will be more pronounced for datasets where there is a big difference in the covariate distribution between the treated and un-treated groups.

## C.2. IV-DID

In this section, we show that when conditioning on the full set of variables $W$, the CLATE can be estimated by the DR-IV learner in (Syrgkanis et al., 2019). By the standard LATE identification argument we can write:

$$\mathbb{E}[Y_1(1) - Y_1(0) \mid D_1(1) > D_1(0), Z = 1, W] = \frac{\mathbb{E}[(Y_1(1) - Y_1(0)) \, 1\{D_1(1) > D_1(0)\} \mid Z = 1, W]}{\mathbb{P}(D_1(1) > D_1(0) \mid Z = 1, W)}$$

$$= \frac{\mathbb{E}[(Y_1(D_1(1)) - Y_1(D_1(0))) \, 1\{D_1(1) > D_1(0)\} \mid Z = 1, W]}{\mathbb{E}[D_1(1) - D_1(0) \mid Z = 1, W]}$$

$$= \frac{\mathbb{E}[Y_1(D_1(1)) - Y_1(D_1(0)) \mid Z = 1, W]}{\mathbb{E}[D_1(1) - D_1(0) \mid Z = 1, W]}$$

Under the parallel trends assumption in the outcome, the numerator is identified as:

$$\mathbb{E}[Y_1(D_1(1)) - Y_1(D_1(0)) \mid Z = 1, W] = \mathbb{E}[Y_1 - Y_0 \mid Z = 1, W] - \mathbb{E}[Y_1(D(0)) - Y_0 \mid Z = 1, W]$$

$$= \mathbb{E}[Y_1 - Y_0 \mid Z = 1, W] - \mathbb{E}[Y_1 - Y_0 \mid Z = 0, W]$$

Moreover, under the parallel trends assumption in the treatment, the denominator is identified as:

$$\mathbb{E}[D_1(1) - D_1(0) \mid Z = 1, W] = \mathbb{E}[D_1 - D_0 \mid Z = 1, W] - \mathbb{E}[D_1(0) - D_0 \mid Z = 1, W]$$

$$= \mathbb{E}[D_1 - D_0 \mid Z = 1, W] - \mathbb{E}[D_1 - D_0 \mid Z = 1, W]$$

For brevity of notation, let $Y$ denote $Y_1 - Y_0$ and let $D$ denote $D_1 - D_0$ Thus, under the PTA assumptions the effect is identified as:

$$\theta_0(W) = \frac{\mathbb{E}[Y \mid Z = 1, W] - \mathbb{E}[Y \mid Z = 0, W]}{\mathbb{E}[D \mid Z = 1, W] - \mathbb{E}[D \mid Z = 0, W]}$$

Moreover, note that we can also write these quantities as conditional covariances.

In particular, let $\alpha(W) = \mathbb{E}[Y \mid Z = 1, W] - \mathbb{E}[Y \mid Z = 0, W]$ and $\gamma(W) = \mathbb{E}[Y \mid Z = 0, W]$. Without loss of generality we can write:

$$\mathbb{E}[Y \mid Z, W] = Z(\mathbb{E}[Y \mid Z = 1, W] - \mathbb{E}[Y \mid Z = 0, W]) + \mathbb{E}[Y \mid Z = 0, W] = Z\alpha(W) + \gamma(W)$$

Thus we can write:

$$Y = Z\alpha(W) + \gamma(W) + \epsilon, \qquad\qquad \mathbb{E}[\epsilon \mid Z, W] = 0$$

Then we have:

$$\mathrm{Cov}(Y, Z \mid W) = \mathbb{E}[\tilde{Y}\tilde{Z} \mid W] = \mathbb{E}[Y\tilde{Z} \mid W]$$

where $\tilde{Y} = Y - \mathbb{E}[Y \mid W]$ and $\tilde{Z} = Z - \pi_0(W) = Z - \mathbb{E}[Z \mid W]$.

Moreover, note that:

$$\mathbb{E}[Y\tilde{Z} \mid W] = \mathbb{E}[\alpha(W)Z\tilde{Z}] + \mathbb{E}[\gamma(W)\tilde{Z} \mid W] + \mathbb{E}[\epsilon\tilde{Z} \mid W]$$

$$= \alpha(W)\mathrm{Var}(Z \mid W) + \gamma(W)\mathbb{E}[\tilde{Z} \mid W] + \mathbb{E}[\mathbb{E}[\epsilon \mid Z, W]\tilde{Z} \mid W]$$

$$= \alpha(W)\mathrm{Var}(Z \mid W)$$

Thus we have:

$$\mathrm{Cov}(Y, Z \mid W) = \mathbb{E}[Y\tilde{Z} \mid W] = (\mathbb{E}[Y \mid Z = 1, W] - \mathbb{E}[Y \mid Z = 0, W]) \, \mathrm{Var}(Z \mid W)$$

Similarly, we can derive:

$$\text{Cov}(D, Z \mid W) = \mathbb{E}[D\tilde{Z} \mid W] = (\mathbb{E}[D \mid Z = 1, W] - \mathbb{E}[D \mid Z = 0, W]) \text{Var}(Z \mid W)$$

Thus we have deduced that we can equivalently identify the conditional LATE among the exposed as:

$$
\begin{aligned}
\theta_0(W) &= \frac{\mathbb{E}[Y\tilde{Z} \mid W]}{\mathbb{E}[D\tilde{Z} \mid W]} = \frac{\text{Cov}(Y, Z \mid W)}{\text{Cov}(D, Z \mid W)} = \frac{(\mathbb{E}[Y \mid Z = 1, W] - \mathbb{E}[Y \mid Z = 0, W]) \text{Var}(Z \mid W)}{(\mathbb{E}[D \mid Z = 1, W] - \mathbb{E}[D \mid Z = 0, W]) \text{Var}(Z \mid W)} \\
&= \frac{\mathbb{E}[Y \mid Z = 1, W] - \mathbb{E}[Y \mid Z = 0, W]}{\mathbb{E}[D \mid Z = 1, W] - \mathbb{E}[D \mid Z = 0, W]}
\end{aligned}
$$

Let $\hat{\alpha}$ be an estimate of:

$$a_0(W) := \mathbb{E}[Y\tilde{Z} \mid W] = (\mathbb{E}[Y \mid Z = 1, W] - \mathbb{E}[Y \mid Z = 0, W]) \text{Var}(Z \mid W)$$

and $\hat{\beta}$ an estimate of:

$$\beta_0(W) := \mathbb{E}[D\tilde{Z} \mid W] = (\mathbb{E}[D \mid Z = 1, W] - \mathbb{E}[D \mid Z = 0, W]) \text{Var}(Z \mid W)$$

and let $\hat{\theta} = \hat{\alpha}/\hat{\beta}$. Then we can construct the random variable

$$\hat{Y}(\hat{g}) = \hat{\theta}(W) + \frac{(Y - \hat{\theta}(W)D)\tilde{Z}}{\hat{\beta}(W)}$$

and the moment equation for the conditional LATT is:

$$\phi = \mathbb{E}[\hat{Y}(\hat{g}) - \theta(W)|W]$$

Similar to the standard DiD case, projecting onto a lower dimensional subset of covariates will give a biased estimated of the CLATE.

## D. Proofs

### D.1. Identification

*Proof of Proposition 2.3.*

$$
\begin{aligned}
&\mathbb{E}[Y_1(1) - Y_1(0)|D = 1, X] \\
={}& \mathbb{E}[Y_1(1) - Y_0(1)|D = 1, X] - \mathbb{E}[Y_1(0) - Y_0(1)|D = 1, X] \\
={}& \mathbb{E}[Y_1 - Y_0|D = 1, X] - \mathbb{E}[Y_1(0) - Y_0(0)|D = 1, X] && \text{(By Assumption 2.2)} \\
={}& \mathbb{E}[Y_1 - Y_0|D = 1, X] - \mathbb{E}\{\mathbb{E}[Y_1(0) - Y_0(0)|D = 1, W]|D = 1, X\} \\
={}& \mathbb{E}[Y_1 - Y_0|D = 1, X] - \mathbb{E}\{\mathbb{E}[Y_1(0) - Y_0(0)|D = 0, W]|D = 1, X\} && \text{(By Assumption 2.1)} \\
={}& \mathbb{E}[Y_1 - Y_0 - \mathbb{E}[Y_1(0) - Y_0(0)|D = 0, W]|D = 1, X]
\end{aligned}
$$

$\square$

*Proof of Proposition A.9.* Here we want to show that the CLATE, $\theta_0(X)$, can be identified as:

$$\theta_0(X) = \frac{\mathbb{E}[Y_1 - Y_0 - \mathbb{E}[Y_1 - Y_0 \mid Z = 0, W] \mid Z = 1, X]}{\mathbb{E}[D_1 - D_0 - \mathbb{E}[D_1 - D_0 \mid Z = 0, W] \mid Z = 1, X]}$$

We first analyze the denominator:

$$\begin{aligned}
&\mathbb{E}[D_1 - D_0 - \mathbb{E}[D_1 - D_0 \mid Z = 0, X] \mid Z = 1, X]\\
&= \mathbb{E}[D_1(1) - D_0(1) - \mathbb{E}[D_1(0) - D_0(0) \mid Z = 0, W] \mid Z = 1, X]\\
&= \mathbb{E}[D_1(1) - D_1(0) + D_1(0) - D_0(1) - \mathbb{E}[D_1(0) - D_0(0) \mid Z = 0, W] \mid Z = 1, X]\\
&= \mathbb{E}[D_1(1) - D_1(0) + D_1(0) - D_0(0) - \mathbb{E}[D_1(0) - D_0(0) \mid Z = 0, W] \mid Z = 1, X] \qquad \text{(By Assumption A.4)}\\
&= \mathbb{E}[D_1(1) - D_1(0) + \mathbb{E}[D_1(0) - D_0(0)|Z = 1, W] - \mathbb{E}[D_1(0) - D_0(0) \mid Z = 0, W] \mid Z = 1, X]\\
&= \mathbb{E}[D_1(1) - D_1(0) \mid Z = 1, X] \qquad \text{(By Assumption A.5)}\\
&= \mathbb{P}(D_1(1) > D_1(0)|Z = 1, X) \qquad \text{(By Assumption A.3)}
\end{aligned}$$

Now we analyze the numerator:

$$\begin{aligned}
&\mathbb{E}[Y_1 - Y_0 - \mathbb{E}[Y_1 - Y_0 \mid Z = 0, W] \mid Z = 1, X]\\
&= \mathbb{E}[Y_1(D(1)) - Y_0(D(1)) - \mathbb{E}[Y_1(D(0)) - Y_0(D(0)) \mid Z = 0, W] \mid Z = 1, X]\\
&= \mathbb{E}[Y_1(D(1)) - Y_1(D(0)) + Y_1(D(0)) - Y_0(D(0)) - \mathbb{E}[Y_1(D(0)) - Y_0(D(0)) \mid Z = 0, W] \mid Z = 1, X]\\
&\qquad\qquad\qquad\qquad\qquad\qquad\qquad\qquad\qquad\qquad\qquad\qquad\qquad \text{(By Assumption A.4)}\\
&= \mathbb{E}[Y_1(D(1)) - Y_1(D(0)) + \mathbb{E}[Y_1(D(0)) - Y_0(D(0))|Z = 1, W] - \mathbb{E}[Y_1(D(0)) - Y_0(D(0)) \mid Z = 0, W] \mid Z = 1, X]\\
&= \mathbb{E}[Y_1(D(1)) - Y_1(D(0)) \mid Z = 1, X] \qquad \text{(By Assumption A.6)}\\
&= \mathbb{E}[(D(1) - D(0))(Y_1(1) - Y_1(0)) \mid Z = 1, X] \qquad \text{(By Assumption A.3)}\\
&= \mathbb{E}[Y_1(1) - Y_1(0) \mid Z = 1, D(1) > D(0), X]\mathbb{P}(D(1) > D(0)|Z = 1, X)
\end{aligned}$$

Thus, combining them, we get:

$$\frac{\mathbb{E}[Y_1 - Y_0 - \mathbb{E}[Y_1 - Y_0 \mid Z = 0, W] \mid Z = 1, X]}{\mathbb{E}[D_1 - D_0 - \mathbb{E}[D_1 - D_0 \mid Z = 0, W] \mid Z = 1, X]} = \mathbb{E}[Y_1(1) - Y_1(0) \mid Z = 1, D(1) > D(0), X]$$

$$\square$$

### D.2. Orthogonal Moments

*Proof of Lemma 3.3.* First, we show that the true CATT function $\theta_0(X) = \mathbb{E}[Y_1(1) - Y_1(0)|D = 1, X]$ is the solution to the moment:

$$\mathbb{E}\left[m(Z; \theta_0, g_0, \pi_0)|X\right] = \mathbb{E}\left[\left(\frac{D - \pi_0(W)}{(1 - \pi_0(W))}\right)(\Delta Y - g_0(W)) - D\theta_0(X) \,\middle|\, X\right] = 0$$

where $Z = (W, D, Y)$, $\Delta Y = Y_1 - Y_0$, $g_0(W) = \mathbb{E}[\Delta Y|D = 0, W]$, $\pi_0(W) = \mathbb{P}(D = 1|W)$. First, since this moment is conditioned on $X$, we can multiply by any functions of $X$. Thus, we can divide by the propensity with $X$, i.e. $\gamma_0(X) = \mathbb{P}(D = 1|X)$, which is bounded away from zero:

$$\mathbb{E}\left[\left(\frac{D - \pi_0(W)}{(1 - \pi_0(W))}\right)(\Delta Y - g_0(W)) - D\theta_0(X) \,\middle|\, X\right] = 0$$

$$\Updownarrow$$

$$\mathbb{E}\left[\left(\frac{D - \pi_0(W)}{(1 - \pi_0(W))\gamma_0(X)}\right)(\Delta Y - g_0(W)) - \frac{D}{\gamma_0(X)}\theta_0(X) \,\middle|\, X\right] = 0$$

The latter term is $\mathbb{E}\left[\frac{D}{\gamma_0(X)}\theta_0(X)\Big|X\right] = \mathbb{E}[\theta_0(X)|D=1,X] = \theta_0(X)$. Now we consider the first term:

$$\mathbb{E}\left[\left(\frac{D-\pi_0(W)}{(1-\pi_0(W))\gamma_0(X)}\right)(\Delta Y - g_0(W))\Big|X\right]$$

$$= \mathbb{E}\left[\left(\frac{D}{\gamma_0(X)} - \frac{(1-D)\pi_0(W)}{(1-\pi_0(W))\gamma_0(X)}\right)(\Delta Y - g_0(W))\Big|X\right]$$

$$= \mathbb{E}\left[\frac{D}{\gamma_0(X)}(\Delta Y - g_0(W))\Big|X\right] - \mathbb{E}\left[\frac{(1-D)\pi_0(W)}{(1-\pi_0(W))\gamma_0(X)}(\Delta Y - g_0(W))\Big|X\right]$$

$$= \mathbb{E}\left[\Delta Y - g_0(W)\,|\,D=1,X\right] - \mathbb{E}\left[\mathbb{E}\left[\frac{(1-D)\pi_0(W)}{(1-\pi_0(W))\gamma_0(X)}(\Delta Y - g_0(W))\Big|W\right]\Big|X\right]$$

$$= \theta_0(X) - \mathbb{E}\left[\mathbb{E}\left[\frac{\pi_0(W)}{\gamma_0(X)}(\Delta Y - g_0(W))\Big|D=0,W\right]\Big|X\right]$$

$$= \theta_0(X) - \mathbb{E}\left[\frac{\pi_0(W)}{\gamma_0(X)}\mathbb{E}\left[\Delta Y - g_0(W)\,|\,D=0,W\right]\Big|X\right]$$

$$= \theta_0(X)$$

Thus, the moment condition is satisfied for the true CATT $\theta_0(X)$. Now, we show that the moment is Neyman orthogonal with respect to all nuisance functions. It suffices to show that the directional derivative with respect to all the nuisance functions are 0 when evaluated at the true nuisance and target functions. Recall that the directional derivative of a functional $m(Z; f)$ with respect to the function $f(W)$ in the direction of $\Delta f(W)$ is defined as: $\partial_f \mathbb{E}[m(z;f)][\Delta f] = \frac{d}{dt}\mathbb{E}[m(z; f+t\cdot\Delta f)]\big|_{t=0}$.

First, we look the directional derivative with respect to the outcome regression $g(W)$:

$$\partial_g \mathbb{E}[m(Z;\theta,g,\pi)|X][\Delta g]|_{\theta_0,g_0,\pi_0} = \mathbb{E}\left[\frac{D-\pi_0(W)}{1-\pi_0(W)}\Delta g(W)\Big|X\right]$$

$$= \mathbb{E}\left[\mathbb{E}\left[\frac{D-\pi_0(W)}{1-\pi_0(W)}\Big|W\right]\Delta g(W)\Big|X\right]$$

$$= \mathbb{E}\left[\frac{\pi_0(W)-\pi_0(W)}{1-\pi_0(W)}\Delta g(W)\Big|X\right] = 0$$

Now, we look at the the directional derivative with respect to the outcome regression $\pi(W)$:

$$\partial_\pi \mathbb{E}[m(Z;\theta,g,\pi)|X][\Delta\pi]|_{\theta_0,g_0,\pi_0} = \mathbb{E}\left[\left(\frac{-(1-\pi_0(W))\Delta\pi(W)+(D-\pi_0(W))\Delta\pi(W)}{(1-\pi_0(W))^2}\right)(\Delta Y - g_0(W))\Big|X\right]$$

$$= \mathbb{E}\left[\left(\frac{\Delta\pi(W)(D-1)}{(1-\pi_0(W))^2}\right)(\Delta Y - g_0(W))\Big|X\right]$$

$$= \mathbb{E}\left[\mathbb{E}\left[\left(\frac{\Delta\pi(W)(D-1)}{(1-\pi_0(W))^2}\right)(\Delta Y - g_0(W))\Big|W\right]\Big|X\right]$$

$$= \mathbb{E}\left[-\left(\frac{\Delta\pi(W)}{1-\pi_0(W)}\right)\mathbb{E}\left[\Delta Y - g_0(W)\,|\,D=0,W\right]\Big|X\right] = 0$$

Thus, we have shown that this moment is Neyman orthogonal with respect to all nuisances. $\square$

*Proof of Theorem 4.7.* First, we show that the true estimand $\theta_0(X) = \mathbb{E}_{source}[m(Z;g_0)|X]$ satisfies the following conditional moment restriction

$$\mathbb{E}\left[m^{DR}(Z;\theta_0,g_0,\pi_0,\alpha_0)\Big|X\right] = \mathbb{E}\left[E(m(Z;g_0)) - \theta_0(X)) + \frac{(1-E)\pi_0(W)}{1-\pi_0(W)}\alpha_0(W)(Y-g_0(W))\Big|X\right] = 0$$

where $\pi_0(W) = \mathbb{P}(E = 1|W)$ and $\alpha(W)$ is the Riesz representer of $\mathbb{E}_s[m(Z; g)|X]$. Similar to the earlier the proof of Lemma 3.3, we can divide both sides of the moment equation by $\gamma_0(X) = \mathbb{P}(E = 1|X)$ since it is bounded away from 0. So it is equivalent to show:

$$\mathbb{E}\left[\frac{E}{\gamma_0(X)}(m(Z; g_0)) - \theta_0(X)) + \frac{(1 - E)\pi_0(W)}{(1 - \pi_0(W))\gamma_0(X)}\alpha_0(W)(Y - g_0(W))\,\Bigg|\, X\right] = 0$$

First, let's look at the first term:

$$\mathbb{E}\left[\frac{E}{\gamma_0(X)}(m(Z; g_0)) - \theta_0(X))\,\Bigg|\, X\right] = \mathbb{E}\left[(m(Z; g_0)) - \theta_0(X))\,|\, E = 1, X\right] = 0$$

Thus, it remains to show that the second term also has conditional expectation of 0.

$$\mathbb{E}\left[\frac{(1 - E)\pi_0(W)}{(1 - \pi_0(W))\gamma_0(X)}\alpha_0(W)(Y - g_0(W))\,\Bigg|\, X\right]$$

$$= \mathbb{E}\left[\mathbb{E}\left[\frac{(1 - E)\pi_0(W)}{(1 - \pi_0(W))\gamma_0(X)}\alpha_0(W)(Y - g_0(W))\,\Bigg|\, W\right]\,\Bigg|\, X\right]$$

$$= \mathbb{E}\left[\mathbb{E}\left[\frac{\pi_0(W)}{\gamma_0(X)}\alpha_0(W)(Y - g_0(W))\,\Bigg|\, E = 0, W\right]\,\Bigg|\, X\right]$$

$$= \mathbb{E}\left[\frac{\pi_0(W)}{\gamma_0(X)}\alpha_0(W)\mathbb{E}\left[(Y - g_0(W))\,|\, E = 0, W\right]\,\Bigg|\, X\right] = 0$$

Now, we proceed to show that the moment $m^{DR}(Z; \theta, g, \pi, \alpha)$ is Neyman orthogonal. First, we look at the directional derivative with respect to the nuisance $g(W)$.

$$\partial_g\mathbb{E}[m^{DR}(Z; \theta, g, \pi, \alpha)|X][\Delta g]|_{\theta_0, g_0, \pi_0, \alpha_0}$$

$$= \partial_g\mathbb{E}[Em(Z; g)|X][\Delta g]|_{g_0} - \mathbb{E}\left[\frac{(1 - E)\pi_0(W)}{1 - \pi_0(W)}\alpha_0(W)\Delta g(W)\,\Bigg|\, X\right]$$

We first look at the first term:

$$\partial_g\mathbb{E}[Em(Z; g)|X][\Delta g]\Big|_{g_0} = \partial_g\mathbb{E}[\gamma_0(W)\mathbb{E}[m(Z; g)|E = 1, X]|X][\Delta g]\Big|_{g_0}$$

$$= \partial_g\mathbb{E}[\gamma_0(X)\mathbb{E}[\alpha_0(W)g(W)|E = 1, X]|X][\Delta g]\Big|_{g_0} \qquad \text{(By the Definition of } \alpha_0(W)\text{)}$$

$$= \mathbb{E}[\gamma_0(X)\mathbb{E}[\alpha_0(W)\Delta g(W)|E = 1, X]|X]$$

$$= \mathbb{E}[E\alpha_0(W)\Delta g(W)|W]$$

Putting this back, we get:

$$\partial_g\mathbb{E}[m^{DR}(Z; \theta, g, \pi)|X][\Delta g]|_{\theta_0, g_0, \pi_0, \alpha_0}$$

$$= \mathbb{E}\left[E\alpha_0(W)\Delta g(W) - \frac{(1 - E)\pi_0(W)}{1 - \pi_0(W)}\alpha_0(W)\Delta g(W)\,\Bigg|\, X\right]$$

$$= \mathbb{E}\left[\alpha_0(W)\Delta g(W)\left(E - \frac{(1 - E)\pi_0(W)}{1 - \pi_0(W)}\right)\,\Bigg|\, X\right]$$

$$= \mathbb{E}\left[\alpha_0(W)\Delta g(W)\mathbb{E}\left[E - \frac{(1 - E)\pi_0(W)}{1 - \pi_0(W)}\,\Bigg|\, W\right]\,\Bigg|\, X\right] = 0$$

Next, we look at the derivative with respect to $\pi(W)$:

$$\partial_\pi \mathbb{E}[m^{DR}(Z;\theta,g,\pi,\alpha)|X][\Delta\pi]|_{\theta_0,g_0,\pi_0,\alpha_0}$$

$$= \mathbb{E}\left[\left(\frac{(1-E)(1-\pi(W))\Delta\pi(W) + (1-E)\pi(W)\Delta\pi(W)}{(1-\pi_0(W))^2}\right)\alpha_0(W)(Y-g_0(W))\,\middle|\, X\right]$$

$$= \mathbb{E}\left[\frac{(1-E)\Delta\pi(W)}{(1-\pi_0(W))^2}\alpha_0(W)(Y-g_0(W))\,\middle|\, X\right]$$

$$= \mathbb{E}\left[\mathbb{E}\left[\frac{(1-E)\Delta\pi(W)}{(1-\pi_0(W))^2}\alpha_0(W)(Y-g_0(W))\,\middle|\, W\right]\,\middle|\, X\right]$$

$$= \mathbb{E}\left[\mathbb{E}\left[\frac{\Delta\pi(W)}{1-\pi_0(W)}\alpha_0(W)(Y-g_0(W))\,\middle|\, E=0,W\right]\,\middle|\, X\right]$$

$$= \mathbb{E}\left[\frac{\Delta\pi(W)}{1-\pi_0(W)}\alpha_0(W)\mathbb{E}\left[(Y-g_0(W)\,|\,E=0,W]\,\middle|\, X\right] = 0$$

Lastly, we show that the directional derivative with respect to $\alpha(W)$ is equal to 0.

$$\partial_\alpha \mathbb{E}[m^{DR}(Z;\theta,g,\pi,\alpha)|X][\Delta\alpha]|_{\theta_0,g_0,\pi_0,\alpha_0}$$

$$= \mathbb{E}\left[\frac{(1-E)\pi_0(W)}{1-\pi_0(W)}\Delta\alpha(W)(Y-g_0(W))\,\middle|\, X\right]$$

$$= \mathbb{E}\left[\mathbb{E}\left[\frac{(1-E)\pi_0(W)}{1-\pi_0(W)}\Delta\alpha(W)(Y-g_0(W))\,\middle|\, W\right]\,\middle|\, X\right]$$

$$= \mathbb{E}\left[\mathbb{E}\left[\pi_0(W)\Delta\alpha(W)(Y-g_0(W))\,|\,E=0,W]\,|\,X\right]$$

$$= \mathbb{E}\left[\pi_0(W)\Delta\alpha(W)\mathbb{E}\left[Y-g_0(W)\,|\,E=0,W]\,|\,X\right] = 0$$

$\square$

*Proof of Lemma A.10.* First we show that the true CLATE, $\theta_0(X) = \mathbb{E}[Y_1(1)-Y_1(0)\,|\,Z=1,D(1)>D(0),X]$, is the solution to the following moment equation:

$$\mathbb{E}\left[m^{DR}(Z;\theta_0,g_{0,Y},g_{0,D},\pi_0)|X\right] = \mathbb{E}\left[\widehat{Z}\{(\Delta Y - g_{0,Y}(W)) - (\Delta D - g_{0,D}(W))\theta(X)\}\,\middle|\, X\right] = 0$$

where $\Delta S = S_1 - S_0$ for $S = Y$ or $D$, $g_{0,S}(W) = \mathbb{E}[S_1-S_0|Z=0,W]$, and $\widehat{Z} = \frac{Z-\pi_0(W)}{1-\pi_0(W)}$ with $\pi_0(W) = \mathbb{P}(Z=1|W)$. We can apply same trick as in the other orthogonality proofs to divide by $\gamma_0(X) = \mathbb{P}(Z=1|X)$. We first consider the first term:

$$\mathbb{E}\left[\frac{\widehat{Z}}{\gamma_0(X)}(\Delta Y - g_{0,Y}(W))\,\middle|\, X\right]$$

$$= \mathbb{E}\left[\frac{Z-\pi_0(W)}{(1-\pi_0(W))\gamma_0(X)}(\Delta Y - g_{0,Y}(W))\,\middle|\, X\right]$$

$$= \mathbb{E}\left[\left(\frac{Z}{\gamma_0(X)} - \frac{(1-Z)\pi_0(W)}{(1-\pi_0(W))\gamma_0(X)}\right)(\Delta Y - g_{0,Y}(W))\,\middle|\, X\right]$$

$$= \mathbb{E}\left[\left(\frac{Z}{\gamma_0(X)}\right)(\Delta Y - g_{0,Y}(W))\,\middle|\, X\right] - \mathbb{E}\left[\frac{\pi_0(W)}{\gamma_0(X)}\mathbb{E}\left[\left(\frac{1-Z}{(1-\pi_0(W))}\right)(\Delta Y - g_{0,Y}(W))\,\middle|\, W\right]\,\middle|\, X\right]$$

$$= \mathbb{E}\left[\Delta Y - g_{0,Y}(W)\,|\,Z=1,X] - \mathbb{E}\left[\frac{\pi_0(W)}{\gamma_0(X)}\mathbb{E}\left[(\Delta Y - g_{0,Y}(W))\,|\,Z=0,W]\,\middle|\, X\right]$$

$$= \mathbb{E}\left[\Delta Y - g_{0,Y}(W)\,|\,Z=1,X]$$

Similarly, for the second term:

$$\mathbb{E}\left[\frac{\widehat{Z}}{\gamma_0(X)}(\Delta D - g_{0,D}(W))\theta(X)\,\middle|\,X\right] = \theta_0(X)\mathbb{E}\left[\frac{\widehat{Z}}{\gamma_0(X)}(\Delta D - g_{0,D}(W))\,\middle|\,X\right]$$
$$= \theta_0(X)\mathbb{E}\left[\Delta D - g_{0,D}(W)\,|\,Z=1,X\right]$$

By the definition of $\theta_0(X)$, this shows that it is a solution to the doubly robust moment equation. Now, we proceed to show that the moment $m^{DR}(Z;\theta,g_Y,g_D,\pi)$ is Neyman orthogonal. First, we look at the directional derivative with respect to the nuisance $g_Y(W)$.

$$\partial_{g_Y}\mathbb{E}[m^{DR}(Z;\theta,g_Y,g_D,\pi)|X][\Delta g_Y]\Big|_{\theta_0,g_{0,Y},g_{0,D},\pi_0} = -E\left[\widehat{Z}\Delta g_Y(W)\,\middle|\,X\right]$$
$$= -\mathbb{E}\left[\frac{Z-\pi_0(W)}{1-\pi_0(W)}\Delta g_Y(W)\,\middle|\,X\right]$$
$$= -\mathbb{E}\left[\mathbb{E}\left[\frac{Z-\pi_0(W)}{1-\pi_0(W)}\,\middle|\,W\right]\Delta g_Y(W)\,\middle|\,X\right] = 0$$

Similarly,

$$\partial_{g_D}\mathbb{E}[m^{DR}(Z;\theta,g_Y,g_D,\pi)|X][\Delta g_D]\Big|_{\theta_0,g_{0,Y},g_{0,D},\pi_0} = E\left[\widehat{Z}\Delta g_D(W)\theta_0(X)\,\middle|\,X\right]$$
$$= \theta_0(X)\mathbb{E}\left[\mathbb{E}\left[\frac{Z-\pi_0(W)}{1-\pi_0(W)}\,\middle|\,W\right]\Delta g_D(W)\,\middle|\,X\right] = 0$$

Lastly, we check the directional derivative with respect to $\pi(W)$:

$$\partial_{\pi}\mathbb{E}[m^{DR}(Z;\theta,g_Y,g_D,\pi)|X][\Delta\pi]\Big|_{\theta_0,g_{0,Y},g_{0,D},\pi_0}$$
$$= \mathbb{E}\left[\frac{-(1-\pi_0(W)\Delta\pi(W)+(Z-\pi_0(W)\Delta\pi(W)}{(1-\pi_0(W))^2}\{(\Delta Y - g_{0,Y}(W))-(\Delta D - g_{0,D}(W))\theta(X)\}\,\middle|\,X\right]$$
$$= \mathbb{E}\left[\frac{(Z-1)\Delta\pi(W)}{(1-\pi_0(W))^2}\{(\Delta Y - g_{0,Y}(W))-(\Delta D - g_{0,D}(W))\theta(X)\}\,\middle|\,X\right]$$
$$= \mathbb{E}\left[\mathbb{E}\left[\frac{(Z-1)\Delta\pi(W)}{(1-\pi_0(W))^2}\{(\Delta Y - g_{0,Y}(W))-(\Delta D - g_{0,D}(W))\theta(X)\}\,\middle|\,W\right]\,\middle|\,X\right]$$
$$= \mathbb{E}\left[\mathbb{E}\left[\frac{(\Delta\pi(W)}{1-\pi_0(W)}\{(\Delta Y - g_{0,Y}(W))-(\Delta D - g_{0,D}(W))\theta(X)\}\,\middle|\,Z=0,W\right]\,\middle|\,X\right]$$
$$= \mathbb{E}\left[\frac{(\Delta\pi(W)}{1-\pi_0(W)}\{E\left[\Delta Y - g_{0,Y}(W)|Z=0,W\right]-\mathbb{E}\left[\Delta D - g_{0,D}(W)|Z=0,W\right]\theta(X)\}\,\middle|\,X\right] = 0$$

$\square$

### D.3. Losses

*Proof of Proposition 3.5.* Note that the true CATT $\theta_0$ satisfies the conditional moment restrictions in Lemma 3.3, which imply that:

$$\mathbb{E}[D\theta_0(X)\,|\,X] = \mathbb{E}[\hat{Y}\,|\,X]$$

Hence, the loss $\mathcal{L}(\theta;\pi_0,g_0)$ at any function $\theta$ can be simplified as:

$$\mathcal{L}(\theta;\pi_0,g_0) = \mathbb{E}\left[D\theta(X)^2 - 2\widehat{Y}\theta(X)\right]$$
$$= \mathbb{E}\left[D\theta(X)^2 - 2\mathbb{E}[\widehat{Y}\,|\,X]\theta(X)\right]$$
$$= \mathbb{E}\left[D\theta(X)^2 - 2\mathbb{E}[D\theta_0(X)\,|\,X]\theta(X)\right]$$
$$= \mathbb{E}\left[D\theta(X)^2 - 2D\theta_0(X)\theta(X)\right]$$

Note that when the loss is evaluated at $\theta_0$, then it takes the value $\mathbb{E}[-D\theta_0(X)^2]$. Moreover, note that minimizing $\mathcal{L}(\theta; \pi_0, g_0)$ is equivalent to minimizing the difference $\mathcal{L}(\theta; \pi_0, g_0) - \mathcal{L}(\theta_0; \pi_0, g_0)$, which in turn simplifies to:

$$\mathbb{E}\left[D\theta(X)^2 - 2D\theta_0(X)\theta(X) + D\theta_0(X)^2\right] = \mathbb{E}\left[D(\theta(X) - \theta_0(X))^2\right]$$

Hence, minimizing $\mathcal{L}(\theta; \pi_0, g_0)$ over any space $\Theta$ is equivalent to minimizing over $\Theta$ the loss function:

$$\mathbb{E}\left[(\theta(X) - \theta_0(X))^2 \mid D = 1\right]$$

$\square$

*Proof of Proposition A.11.* Note that the true CATT $\theta_0$ satisfies the conditional moment restrictions in Lemma A.10, which imply that:

$$\mathbb{E}[\widehat{Z}(\Delta D - g_D(W))\theta_0(X) \mid X] = \mathbb{E}[\widehat{Z}(\Delta Y - g_Y(W)) \mid X]$$

Let $\eta_0$ denote the set of nuisance functions. The loss $\mathcal{L}_{IV}(\theta; \eta_0)$ at any function $\theta$ can be simplified as:

$$\begin{aligned}
\mathcal{L}_{IV}(\theta; \eta_0) &= \mathbb{E}\left[\widehat{Z}(\Delta D - g_D(W))\theta(X)^2 - 2\widehat{Z}(\Delta D - g_D(W))\theta_0(X)\theta(X)\right] \\
&= \mathbb{E}\left[\widehat{Z}(\Delta D - g_D(W))\theta(X)^2 - 2\mathbb{E}[\widehat{Z}(\Delta D - g_D(W))\theta_0(X) \mid X]\theta(X)\right] \\
&= \mathbb{E}\left[\widehat{Z}(\Delta D - g_D(W))\left(\theta(X)^2 - 2\theta_0(X)\theta(X)\right)\right]
\end{aligned}$$

Note that when the loss is evaluated at $\theta_0$, then it takes the value $\mathbb{E}[-\widehat{Z}(\Delta D - g_D(W))\theta_0(X)^2]$. Moreover, note that minimizing $\mathcal{L}_{IV}(\theta; \eta_0)$ is equivalent to minimizing the difference $\mathcal{L}_{IV}(\theta; \eta_0) - \mathcal{L}_{IV}(\theta_0; \eta_0)$, which in turn simplifies to:

$$\begin{aligned}
&\mathbb{E}\left[\widehat{Z}(\Delta D - g_D(W))\left(\theta(X)^2 - 2\theta_0(X)\theta(X) + \theta_0(X)^2\right)\right] \\
&= \mathbb{E}\left[\widehat{Z}(\Delta D - g_D(W))(\theta(X) - \theta_0(X))^2\right] \\
&= \mathbb{E}\left[\left(Z - \frac{(1-Z)\pi_0(W)}{1 - \pi_0(W)}\right)(\Delta D - g_D(W))(\theta(X) - \theta_0(X))^2\right] \\
&= \mathbb{E}\left[Z(\Delta D - g_D(W))(\theta(X) - \theta_0(X))^2\right] - \mathbb{E}\left[\pi_0(W)(\theta(X) - \theta_0(X))^2\mathbb{E}\left[(\Delta D - g_D(W)) \mid Z = 0, W\right]\right] \\
&= \mathbb{E}\left[Z(\Delta D - g_D(W))(\theta(X) - \theta_0(X))^2\right] \\
&= \mathbb{E}\left[(\Delta D - g_D(W))(\theta(X) - \theta_0(X))^2 \mid Z = 1\right]\mathbb{P}(Z = 1) \\
&= \mathbb{E}\left[\mathbb{E}\left[(\Delta D - g_D(W)) \mid Z = 1, X\right](\theta(X) - \theta_0(X))^2 \mid Z = 1\right]\mathbb{P}(Z = 1) \\
&= \mathbb{E}\left[\mathbb{P}(D_1(1) > D_1(0) \mid Z = 1, X)(\theta(X) - \theta_0(X))^2 \mid Z = 1\right]\mathbb{P}(Z = 1) \qquad \text{(See the proof of Proposition A.9)} \\
&= \mathbb{E}\left[(D_1(1) > D_1(0))(\theta(X) - \theta_0(X))^2 \mid Z = 1\right]\mathbb{P}(Z = 1) \\
&= \mathbb{E}\left[(\theta(X) - \theta_0(X))^2 \mid Z = 1, D(1) > D(0)\right]\mathbb{P}(Z = 1)\mathbb{P}(D(1) > D(0) \mid Z = 1)
\end{aligned}$$

Hence, minimizing $\mathcal{L}_{IV}(\theta; \eta_0)$ over any space $\Theta$ is equivalent to minimizing over $\Theta$ the loss function:

$$\mathbb{E}\left[(\theta(X) - \theta_0(X))^2 \mid Z = 1, D(1) > D(0)\right]$$

$\square$

### D.4. Rates

Before proving Theorem 3.6, we first present some auxiliary Lemmas.

**Lemma D.1.** *Let* $\eta = (\pi, g)$ *denote the set of nuisance functions, and let* $\eta_0$ *be the true nuisance functions. Consider the loss defined in Proposition 3.5. Then, we have that for all* $\theta_1$, $\theta_2$, $\eta_1$ *and* $\eta_2$,

$$|\mathcal{L}(\theta_1; \eta_1) - \mathcal{L}(\theta_2; \eta_1) - \mathcal{L}(\theta_2; \eta_1) + \mathcal{L}(\theta_2; \eta_2)| \leq 2\sqrt{\mathbb{E}\left[\mathbb{E}\left[\widehat{Y}(\eta_1) - \widehat{Y}(\eta_2)\Big|X\right]^2\right]}\|\theta_1 - \theta_2\|$$

.

*Proof of Lemma D.1.*

$$\begin{aligned}
&|\mathcal{L}(\theta_1; \eta_1) - \mathcal{L}(\theta_2; \eta_1) - \mathcal{L}(\theta_2; \eta_1) + \mathcal{L}(\theta_2; \eta_2)| \\
&= \left|\mathbb{E}\left[D(\theta_1^2(X) - \theta_2^2(X)) + 2\widehat{Y}(\eta_1)(\theta_2(X) - \theta_1(X))\right] - \mathbb{E}\left[D(\theta_1^2(X) - \theta_2^2(X)) + 2\widehat{Y}(\eta_2)(\theta_2(X) - \theta_1(X))\right]\right| \\
&= \left|\mathbb{E}\left[2\left(\widehat{Y}(\eta_1) - \widehat{Y}(\eta_2)\right)(\theta_1(X) - \theta_2(X))\right]\right| \\
&= 2\left|\mathbb{E}\left[\mathbb{E}\left[\left(\widehat{Y}(\eta_1) - \widehat{Y}(\eta_2)\right)(\theta_1(X) - \theta_2(X))\Big|X\right]\right]\right| \\
&\leq 2\sqrt{\mathbb{E}\left[\mathbb{E}\left[\widehat{Y}(\eta_1) - \widehat{Y}(\eta_2)\Big|X\right]^2\right]}\|\theta_1(X) - \theta_2(X)\|
\end{aligned}$$

$\square$

We then show that the bias in the pseudo-outcome $\widehat{Y}$ is equal to the product of the biases in the nuisance functions.

**Lemma D.2.** *Let* $\eta = (\pi, g)$ *denote the set of nuisance functions, and let* $\eta_0$ *be the true nuisance functions. Consider the pseudo-outcome defined in Proposition 3.5. Then we have:*

$$\mathbb{E}[\widehat{Y}(\eta_0) - \widehat{Y}(\hat{\eta})|X] = \mathbb{E}\left[(\hat{g}(W) - g_0(W))\frac{\pi_0(W) - \hat{\pi}(W)}{1 - \hat{\pi}(W)}\Big|X\right]$$

*Proof of Lemma D.2.*

$$\begin{aligned}
&\mathbb{E}[\widehat{Y}(\eta_0) - \widehat{Y}(\hat{\eta})|X] \\
&= \mathbb{E}\left[\frac{D - \pi_0(W)}{1 - \pi_0(W)}(\Delta Y - g_0(W)) - \frac{D - \hat{\pi}(W)}{1 - \hat{\pi}(W)}(\Delta Y - \hat{g}(W))\Big|X\right] \\
&= \mathbb{E}\left[\mathbb{E}\left[\left(D - \frac{(1-D)\pi_0(W)}{1 - \pi_0(W)}\right)(\Delta Y - g_0(W)) - \left(D - \frac{(1-D)\hat{\pi}(W)}{1 - \hat{\pi}(W)}\right)(\Delta Y - \hat{g}(W))\Big|W\right]\Big|X\right] \\
&= \mathbb{E}\left[\mathbb{E}\left[D(\hat{g}(W) - g_0(W)) - \frac{(1-D)\pi_0(W)}{1 - \pi_0(W)}(\Delta Y - g_0(W)) + \frac{(1-D)\hat{\pi}(W)}{1 - \hat{\pi}(W)}(\Delta Y - \hat{g}(W))\Big|W\right]\Big|X\right] \\
&= \mathbb{E}\left[\mathbb{E}[\pi_0(W)(\hat{g}(W) - g_0(W))\,|\,W] - \mathbb{E}[\Delta Y - g_0(W)\,|\,D=0, W] + \mathbb{E}\left[\frac{(1-D)\hat{\pi}(W)}{1 - \hat{\pi}(W)}(\Delta Y - \hat{g}(W))\Big|W\right]\Big|X\right] \\
&= \mathbb{E}\left[\pi_0(W)(\hat{g}(W) - g_0(W)) + \mathbb{E}\left[\frac{1-D}{1 - \pi_0(W)}\frac{(1-\pi_0(W))\hat{\pi}(W)}{1 - \hat{\pi}(W)}(\Delta Y - \hat{g}(W))\Big|W\right]\Big|X\right] \\
&= \mathbb{E}\left[\pi_0(W)(\hat{g}(W) - g_0(W)) + \mathbb{E}\left[\frac{(1-\pi_0(W))\hat{\pi}(W)}{1 - \hat{\pi}(W)}(\Delta Y - \hat{g}(W))\Big|D=0, W\right]\Big|X\right] \\
&= \mathbb{E}\left[\pi_0(W)(\hat{g}(W) - g_0(W)) + \mathbb{E}\left[\frac{(1-\pi_0(W))\hat{\pi}(W)}{1 - \hat{\pi}(W)}(g_0(W) - \hat{g}(W))\Big|D=0, W\right]\Big|X\right] \\
&= \mathbb{E}\left[(\hat{g}(W) - g_0(W))\frac{\pi_0(W) - \hat{\pi}(W)}{1 - \hat{\pi}(W)}\Big|X\right]
\end{aligned}$$

$\square$

The rates in Theorem 3.6 is an application of Theorem 1 in Foster & Syrgkanis, 2023. We reproduce the theorem in our notation for completeness. Let $d(\hat{\eta}, \eta_0)$ denote a distance metric for the function space of the nuisance functions $\mathcal{F}$, and $\|(\cdot)\|_\Theta$ denote a norm for $\Theta$. We denote $\text{Star}(\Theta, \theta)$ to be the star hull, i.e.$\text{Star}(\Theta, \theta) = \{t\theta + (1 - t)\theta' \,|\, \forall \theta' \in \Theta, t \in [0, 1]\}$. Moreover, let $\theta'$ be an arbitrary element in $\Theta$.

**Assumption D.3** (First Order Optimality). $\theta'$ satisfies the first-order optimality condition for $\mathcal{L}(\theta; \eta_0)$:

$$\partial_\theta \mathcal{L}(\theta; \eta_0)[\theta - \theta'] \geq 0 \quad \forall \quad \theta \in \text{Star}(\Theta, \theta')$$

**Assumption D.4** (Higher Order Smoothness). There exist constant $\beta_1$ such that:

$$\partial_\theta^2 \mathcal{L}(\bar{\theta}, \eta_0)[\theta - \theta', \theta - \theta'] \leq \beta_1 \|\theta - \theta'\|_\Theta^2$$

for all $\theta \in \Theta$ and all $\bar{\theta} \in \text{Star}(\Theta, \theta')$.

**Assumption D.5** (Strong Convexity). The population loss is strongly convex with respect to $\theta$, i.e. there exist constants $\lambda$, $\kappa > 0$ and $r \geq 0$, such that for all $\theta \in \Theta$, $\theta' \in \text{Star}(\Theta, \theta')$, and $\eta \in \mathcal{F}$:

$$\partial_\theta^2 \mathcal{L}(\bar{\theta}, \eta)[\theta - \theta', \theta - \theta'] \geq \lambda \|\theta - \theta'\|^2 - \kappa d(\eta, \eta_0)^{\frac{4}{1+r}}$$

**Assumption D.6.** There exist $r \in [0, 1)$ and constant $\beta_2$ such that for all $\theta, \theta' \in \text{Star}(\Theta, \theta')$ and all $\eta_1, \eta_2$ in $\mathcal{F}$:

$$\|\mathcal{L}(\theta; \eta_1) - \mathcal{L}(\theta'; \eta_1) - \mathcal{L}(\theta; \eta_2) + \mathcal{L}(\theta'; \eta_2)\| \leq \beta_2 \|\theta - \theta'\|_\Theta^{1-r} d(\eta_1, \eta_2)^2$$

**Theorem D.7** (Theorem 1 from (Foster & Syrgkanis, 2023)). *Suppose Assumptions D.3, D.4, D.5, and D.6 are satisfied for some $\theta' \in \Theta$. Then for any $\theta \in \Theta$, the following holds:*

$$\|\theta - \theta'\|_\Theta^2 \leq \frac{4}{\lambda} \left(\mathcal{L}(\theta, \hat{\eta}) - \mathcal{L}(\theta', \hat{\eta})\right) + \left(\left(\frac{\beta_2}{\lambda}\right)^{\frac{2}{1+r}} + \frac{\kappa}{\lambda}\right) d(\eta_0, \hat{\eta})^{\frac{4}{1+r}}$$

We are finally ready to prove Theorem 3.6.

*Proof of Theorem 3.6.* Since results follow from Theorem D.7, we first show that the minimizer of the loss in the function class $\Theta$, i.e. $\theta_*$, satisfies Assumptions D.3, D.4, D.5, and D.6 for the proposed loss $\mathcal{L}(\theta; \eta)$ with $\|(\cdot)\|_\Theta = \|(\cdot)\|_{D=1}$. Assumption D.3 is satisfied when $\Theta$ is convex or when $\theta_0 \in \Theta$. Assumptions D.4 and D.5 requires us to bound:

$$\partial_\theta^2 \mathcal{L}(\bar{\theta}, \hat{\eta})[\theta - \theta_*, \theta - \theta_*] = \mathbb{E}[D(\theta(X) - \theta_*(X))^2] = \rho \|(\theta(X) - \theta_*(X))\|_{D=1}^2$$

Thus Assumptions D.4 and D.5 are satisfied with $\beta_1 = \lambda = \rho$ and $\kappa = 0$. To show Assumption D.6, we need to convert the $\|(\cdot)\|_2$ in D.2 into $\|(\cdot)\|_{D=1}$:

$$\begin{aligned}
\|(\theta(X) - \theta_*(X))\|^2 &= \int (\theta(X) - \theta_*(X))^2 \mathbb{P}p(X)dX \\
&= \int (\theta(X) - \theta_*(X))^2 \mathbb{P}(D = 1|X) \frac{1}{\mathbb{P}(D = 1|X)} p(X)dX \\
&\leq \frac{1}{c} \int (\theta(X) - \theta_*(X))^2 \mathbb{P}(D = 1|X)p(X)dX \\
&= \frac{1}{c} \|\theta(X) - \theta_*(X)\|_\Theta^2
\end{aligned}$$

Thus, Lemmas D.1 and D.2 imply Assumption D.6 with $r = 0$, $\beta_2 = \frac{3}{c}$, and

$$d(\eta, \eta_0)^2 = \mathbb{E}\left[\mathbb{E}\left[(\hat{g}(W) - g_0(W))\left(\frac{\pi_0(W) - \hat{\pi}(W)}{1 - \hat{\pi}(W)}\right) \Big| X\right]^2\right]^{1/2}$$

Thus invoking Theorem D.7, we get that:

$$\|\theta - \theta'\|_\Theta^2 \leq \frac{4}{\rho}R(n,\delta) + \frac{2}{\rho^2 c^2}\mathbb{E}\left[\mathbb{E}\left[(\hat{g}(W) - g_0(W))\left(\frac{\pi_0(W) - \hat{\pi}(W)}{1 - \hat{\pi}(W)}\right)\bigg| X\right]^2\right]$$

$\square$

Analogously, we can prove the rates in the case with instrument. Consider $\mathcal{L}_{IV}(\theta;\eta)$ from Proposition A.11, where we let $\eta$ denote the set of nuisances $\pi(W)$, $g_D(W)$ and $g_Y(W)$. We first present an auxiliary lemma to bound $|\mathcal{L}_{IV}(\theta_1;\eta_1) - \mathcal{L}_{IV}(\theta_2;\eta_1) - (\mathcal{L}_{IV}(\theta_2;\eta_1) - \mathcal{L}_{IV}(\theta_2;\eta_2))|$.

**Lemma D.8.** *Let $\eta = (\pi, g_Y, g_D)$ denote the set of nuisance functions, and let $\eta_0 = (\pi_0, g_{0,Y}, g_{0,D})$ be the true nuisance functions. Consider the loss defined in Proposition A.11. Assume there exist finite constant $B$ such that $|\theta(X)| \leq B$ for all $X$ with positive measure, and all $\theta \in \Theta$. Then, we have that for all $\theta_1$, $\theta_2$, $\eta$,*

$$|\mathcal{L}_{IV}(\theta_1;\eta) - \mathcal{L}_{IV}(\theta_2;\eta) - (\mathcal{L}_{IV}(\theta_2;\eta_0) - \mathcal{L}_{IV}(\theta_2;\eta_0))|$$

$$\leq 4B^2\mathbb{E}\left[\mathbb{E}\left[\left(\frac{\pi_0(W) - \pi(W)}{1 - \pi(W)}\right)(g_{0,D}(W) - g_D(W))\bigg| X\right]^2\right]^{\frac{1}{2}}\|\theta(X) - \theta(X)\|$$

$$+ 2\mathbb{E}\left[\mathbb{E}\left[\left(\frac{\pi_0(W) - \pi(W)}{1 - \pi(W)}\right)(g_{0,Y}(W) - g_Y(W))\bigg| X\right]^2\right]^{\frac{1}{2}}\|\theta(X) - \theta(X)\|$$

*Proof of Lemma D.8.*

$$\mathcal{L}_{IV}(\theta_1;\eta) - \mathcal{L}_{IV}(\theta_2;\eta) = \mathbb{E}\left[\widehat{Z}(\eta)\left\{(\Delta D - g_D(W))(\theta_1^2(X) - \theta_2^2(X)) - 2(\Delta Y - g_Y(W))(\theta_1(X) - \theta_2(X))\right\}\right]$$

$$\mathcal{L}_{IV}(\theta_1;\eta_0) - \mathcal{L}_{IV}(\theta_2;\eta_0) = \mathbb{E}\left[\widehat{Z}(\eta_0)\left\{(\Delta D - g_{0,D}(W))(\theta_1^2(X) - \theta_2^2(X)) - 2(\Delta Y - g_{0,Y}(W))(\theta_1(X) - \theta_2(X))\right\}\right]$$

Let's first consider the $\mathbb{E}\left[\widehat{Z}(\eta_0)(D - g_{0,D}(W))(\theta_1^2(X) - \theta_2^2(X))\right]$ term:

$$\mathbb{E}\left[\widehat{Z}(\eta_0)(\Delta D - g_{0,D}(W))(\theta_1^2(X) - \theta_2^2(X))\right]$$
$$= E\left[\left(Z - \frac{(1-Z)\pi_0(W)}{1 - \pi_0(W)}\right)(\Delta D - g_{0,D}(W))(\theta_1^2(X) - \theta_2^2(X))\right]$$
$$= \mathbb{E}[Z\Delta D(\theta_1^2(X) - \theta_2^2(X))] - \mathbb{E}[Zg_{0,D}(W)(\theta_1^2(X) - \theta_2^2(X))]$$
$$\quad - E\left[E\left[(\Delta D - g_{0,D}(W))|Z=0, W\right]\pi_0(W)(\theta_1^2(X) - \theta_2^2(X))\right]$$
$$= \mathbb{E}[\Delta DZ(\theta_1^2(X) - \theta_2^2(X))] - \mathbb{E}[Zg_{0,D}(W)(\theta_1^2(X) - \theta_2^2(X))]$$

Now, for the $\mathbb{E}\left[\widehat{Z}(\eta)(D - g_D(W))(\theta_1^2(X) - \theta_2^2(X))\right]$ term:

$$\mathbb{E}\left[\widehat{Z}(\eta)(\Delta D - g_D(W))(\theta_1^2(X) - \theta_2^2(X))\right]$$

$$= \mathbb{E}\left[\left(Z - \frac{(1-Z)\pi(W)}{1-\pi(W)}\right)(\Delta D - g_D(W))(\theta_1^2(X) - \theta_2^2(X))\right]$$

$$= \mathbb{E}[\Delta D Z(\theta_1^2(X) - \theta_2^2(X))] - \mathbb{E}[Z g_D(W)(\theta_1^2(X) - \theta_2^2(X))]$$

$$\quad - \mathbb{E}\left[\mathbb{E}\left[\frac{1-Z}{1-\pi_0(W)}(\Delta D - g_D(W))|W\right]\frac{(1-\pi_0(W))\pi_0(W)}{1-\pi(W)}(\theta_1^2(X) - \theta_2^2(X))\right]$$

$$= \mathbb{E}[\Delta D Z(\theta_1^2(X) - \theta_2^2(X))] - \mathbb{E}[Z g_D(W)(\theta_1^2(X) - \theta_2^2(X))]$$

$$\quad - \mathbb{E}\left[\mathbb{E}\left[(\Delta D - g_D(W))|Z = 0, W\right]\frac{(1-\pi_0(W))\pi_0(W)}{1-\pi(W)}(\theta_1^2(X) - \theta_2^2(X))\right]$$

$$= \mathbb{E}[\Delta D Z(\theta_1^2(X) - \theta_2^2(X))] - \mathbb{E}[Z g_D(W)(\theta_1^2(X) - \theta_2^2(X))]$$

$$\quad - \mathbb{E}\left[\frac{(1-\pi_0(W))\pi_0(W)}{1-\pi(W)}(g_{0,D}(W) - g_D(W))(\theta_1^2(X) - \theta_2^2(X))\right]$$

Putting them together, we get:

$$\mathbb{E}\left[\widehat{Z}(\eta)(\Delta D - g_D(W))(\theta_1^2(X) - \theta_2^2(X)) - \widehat{Z}(\eta_0)(\Delta D - g_{0,D}(W))(\theta_1^2(X) - \theta_2^2(X))\right]$$

$$= \mathbb{E}[Z(g_{0,D}(W) - g_D(W))(\theta_1^2(X) - \theta_2^2(X))] - \mathbb{E}\left[\frac{(1-\pi_0(W))\pi_0(W)}{1-\pi(W)}(g_{0,D}(W) - g_D(W))(\theta_1^2(X) - \theta_2^2(X))\right]$$

$$= \mathbb{E}\left[\left(Z - \frac{(1-\pi_0(W))\pi_0(W)}{1-\pi(W)}\right)(g_{0,D}(W) - g_D(W))(\theta_1^2(X) - \theta_2^2(X))\right]$$

$$= \mathbb{E}\left[\frac{\pi_0(W) - \pi(W)}{1-\pi(W)}(g_{0,D}(W) - g_D(W))(\theta_1^2(X) - \theta_2^2(X))\right]$$

Similarly,

$$\mathbb{E}\left[\widehat{Z}(\eta)(\Delta Y - g_Y(W))(\theta_1(X) - \theta_2(X)) - \widehat{Z}(\eta_0)(\Delta Y - g_{0,Y}(W))(\theta_1(X) - \theta_2(X))\right]$$

$$= \mathbb{E}\left[\frac{\pi_0(W) - \pi(W)}{1-\pi(W)}(g_{0,Y}(W) - g_Y(W))(\theta_1(X) - \theta_2(X))\right]$$

Thus, we have shown that:

$$|\mathcal{L}_{IV}(\theta_1; \eta) - \mathcal{L}_{IV}(\theta_2; \eta) - (\mathcal{L}_{IV}(\theta_2; \eta_0) - \mathcal{L}_{IV}(\theta_2; \eta_0))|$$

$$= \left|\mathbb{E}\left[\frac{\pi_0(W) - \pi(W)}{1-\pi(W)}(g_{0,D}(W) - g_D(W))(\theta_1^2(X) - \theta_2^2(X)) - 2\frac{\pi_0(W) - \pi(W)}{1-\pi(W)}(g_{0,Y}(W) - g_Y(W))(\theta_1(X) - \theta_2(X))\right]\right|$$

$$\leq \mathbb{E}\left[\mathbb{E}\left[\left(\frac{\pi_0(W) - \pi(W)}{1-\pi(W)}\right)(g_{0,D}(W) - g_D(W))\bigg|X\right]^2\right]^{\frac{1}{2}}\mathbb{E}\left[(\theta_1^2(X) - \theta_2^2(X))^2\right]^{\frac{1}{2}}$$

$$\quad + 2\mathbb{E}\left[\mathbb{E}\left[\left(\frac{\pi_0(W) - \pi(W)}{1-\pi(W)}\right)(g_{0,Y}(W) - g_Y(W))\bigg|X\right]^2\right]^{\frac{1}{2}}\mathbb{E}\left[(\theta_1(X) - \theta_2(X))^2\right]^{\frac{1}{2}}$$

$$\leq 4B^2\mathbb{E}\left[\mathbb{E}\left[\left(\frac{\pi_0(W) - \pi(W)}{1-\pi(W)}\right)(g_{0,D}(W) - g_D(W))\bigg|X\right]^2\right]^{\frac{1}{2}}\|\theta(X) - \theta(X)\|$$

$$\quad + 2\mathbb{E}\left[\mathbb{E}\left[\left(\frac{\pi_0(W) - \pi(W)}{1-\pi(W)}\right)(g_{0,Y}(W) - g_Y(W))\bigg|X\right]^2\right]^{\frac{1}{2}}\|\theta(X) - \theta(X)\|$$

□

We can now prove Theorem A.12.

*Proof of Theorem A.12.* Since results follows from Theorem D.7, we first show that the minimizer of the loss in the function class $\Theta$, i.e. $\theta_*$, satisfies Assumptions D.3, D.4, D.5, and D.6 for the proposed loss $\mathcal{L}_{IV}(\theta; \eta)$ with $\|(\cdot)\|_\Theta = \|(\cdot)\|_{Z=1,CM}$ First, Assumption D.3 is satisfied when $\Theta$ is convex or when $\theta_0 \in \Theta$. Now, we look at the second order directional derivative with respect to $\theta$. Following the same steps as in the proof of Porposition A.11, we get:

$$
\begin{aligned}
&\partial_\theta^2 \mathcal{L}_{IV}(\bar{\theta}, \eta_0)[\theta - \theta_*, \theta - \theta_*] \\
&= \mathbb{E}[\widehat{Z}(\eta_0)(\Delta D - g_{0,D}(W))(\theta(X) - \theta_*(X))^2] \\
&= \mathbb{E}\left[(\theta(X) - \theta_*(X))^2 | Z=1, D(1) > D(0)\right] \mathbb{P}(Z=1)\mathbb{P}(D(1) > D(0)|Z=1) \\
&= hk \|\theta(X) - \theta_*(W)\|_{Z=1,CM}
\end{aligned}
$$

Thus Assumption D.4 is statisfied with $\beta_1 = hk$

However, for Assumption D.5, we need to bound the second directional derivative for any $\eta$. Therefore, we consider the distance between $\partial_\theta^2 \mathcal{L}_{IV}(\bar{\theta}, \eta)[\theta - \theta_*, \theta - \theta_*] - \partial_\theta^2 \mathcal{L}_{IV}(\bar{\theta}, \eta_0)[\theta - \theta_*, \theta - \theta_*]$:

$$
\begin{aligned}
&\partial_\theta^2 \mathcal{L}_{IV}(\bar{\theta}, \eta)[\theta - \theta_*, \theta - \theta_*] - \partial_\theta^2 \mathcal{L}_{IV}(\bar{\theta}, \eta_0)[\theta - \theta_*, \theta - \theta_*] \\
&= 2\mathbb{E}[\widehat{Z}(\eta)(\Delta D - g_D(W))(\theta(X) - \theta_*(X))^2] - \mathbb{E}[\widehat{Z}(\eta_0)(\Delta D - g_{0,D}(W))(\theta(X) - \theta_*(X))^2] \\
&= 2\mathbb{E}\left[\frac{\pi_0(W) - \pi(W)}{1 - \pi(W)}(g_{0,D}(W) - g_D(W))(\theta(X) - \theta_*(X))^2\right] \quad \text{(By the same reasoning in the proof of Lemma D.8)} \\
&\leq 2\mathbb{E}\left[\mathbb{E}\left[(\hat{g}_D(W) - g_{0,D}(W))\left(\frac{\pi_0(W) - \hat{\pi}(W)}{1 - \hat{\pi}(W)}\right)\bigg| X\right]^2\right]^{1/2} \|(\theta(X) - \theta_*(X))\|_4^2 \quad \text{(By Cauchy-Schwarz)} \\
&\leq 8B^2 \mathbb{E}\left[\mathbb{E}\left[(\hat{g}_D(W) - g_{0,D}(W))\left(\frac{\pi_0(W) - \hat{\pi}(W)}{1 - \hat{\pi}(W)}\right)\bigg| X\right]^2\right]^{1/2} \|(\theta(X) - \theta_*(X))\|_2^2 \\
&\leq \frac{8B^2}{c} \mathbb{E}\left[\mathbb{E}\left[(\hat{g}_D(W) - g_{0,D}(W))\left(\frac{\pi_0(W) - \hat{\pi}(W)}{1 - \hat{\pi}(W)}\right)\bigg| X\right]^2\right]^{1/2} \|(\theta(X) - \theta_*(X))\|_\Theta^2
\end{aligned}
$$

Thus, for sufficiently small nuisance error, Assumption D.5 is satisfied with $\kappa = 0$, and

$$
\lambda = hk - \frac{8B^2}{c}\text{Error}(\pi, g_D)
$$

Lemma D.8 implies Assumption D.6 with $r = 0$, $\beta_2 = \frac{1}{c}\max\{4B^2, 2\}$, and $d(\eta, \eta_0)^2 = \text{Error}(\pi, \hat{g}_D) + \text{Error}(\pi, \hat{g}_Y)$. Thus invoking Theorem D.7, we get that:

$$
\|\theta(X) - \theta_*(X)\|_\Theta^2 \leq \frac{4}{hk - \frac{8B^2}{c}\text{Error}(\pi, g_D)}R_n^2 + \left(\frac{\max(4B^2, 2)}{c\left(hk - \frac{8B^2}{c}\text{Error}(\pi, g_D)\right)}\right)(\text{Error}(\pi, \hat{g}_Y) + \text{Error}(\pi, \hat{g}_D))
$$

□

# E. Additional Experiment Details and Results

## E.1. Experiment Setup

Here we describe the data generating processes (DGP) for the fully synthetic experimens. We consider soome observed covariates $W$ with dimension $d_W$, and some unobserved confounding $U$, of dimension $d_U$. Let $\mu_W, \mu_U$ be the mean of $W$

and $U$, where each entry is sampled from a uniform ditsribution ranging from 0 to 1. Let $I_d$ denote the identity matrix with dimension $d$.

$$W \sim \mathcal{N}(\mu_W, I_{d_X})$$
$$W_{masked} \sim \text{Half of the dimensions of } W \text{ are randomly set to } 0$$
$$U \sim \mathcal{N}(\mu_U, I_{d_U})$$
$$p = \frac{1}{1 + exp(-\frac{1}{2}\beta_D^T(W - \mu_W) * (\alpha_U^T(U - \mu_U))^2)} \qquad (p \text{ is clipped s.t. } p \in [0.9, 0.1])$$
$$D \sim Binomial(p)$$
$$\theta_0 = \frac{1}{2}W_1 * \mathbb{1}(W_2 > 0)$$

For experiments with DGP that satisfies the conditional parallel trends assumptions:

$$Y_0 = 5(\alpha_U^T(U - \mu_U))^2 W_6 + W_2 + \epsilon_0, \quad \epsilon_0 \sim \mathcal{N}(0, 0.5)$$
$$Y_1 = 5(\alpha_U^T(U - \mu_U))^2 W_6 + \mathbb{1}(W_1 > 0)W_1 + \beta_Y^T W_{masked} + W_3 + D * \theta_0 + \epsilon_1, \quad \epsilon_1 \sim \mathcal{N}(0, 0.5)$$

The results in Table 1 and 4 are generated using this process with $d_W = 20$ and $d_U = 5$. We also ran experiments with higher dimensional covariates ($d_W = 100$), and the results are presented in Table 6. The results in Table 2 is generated using the same setup, but with $0.1 * p$ as the treatment probabilities. These results all showcase that our proposed doubly robust CATT learner out performs the baseline methods. In addition to this DGP, we also experimented with a DGP that does not satisfy the conditional parallel trends assumptions.

$$\gamma \sim Uniform([-1, 1])$$
$$Y_0 = (\alpha_U^T(U - \mu_U))^2 X_6 + X_2 + \epsilon_0, \quad \epsilon_0 \sim \mathcal{N}(0, 0.5)$$
$$m = |Y_0|$$
$$Y_1 = (\alpha_U^T(U - \mu_U))^2 X_6 + m\gamma^T X \odot X + \mathbb{1}(X_1 > 0)X_2 + D * \theta_0 + \epsilon_1, \quad \epsilon_1 \sim \mathcal{N}(0, 0.5)$$

Experiment results for this DGP is presented in Table 7. We see that in this case, the conditional parallel trends is violated so the learner that assumes conditional parallel trends has a higher MSE than the those that assume lagged dependent outcome (as this DGP has a lagged outcome component). Moreover, we see that even when the assumptions are violated, the proposed learner is still more robust than the baseline outcome regression learner.

For the semi-sythetic experiments on the minimum wage dataset, each dataset is constructed by first sampling 10000 units with replacement from the original dataset. We keep the covariate and pre-treatment outcome information, and generate the treatment assignment and the outcome in the post-treatment time period. The probability of receiving treatment is generated from the logitistic transformation of a linear transformation of a linear function of 2 "region" variables that are binary, and the log average payment information for year 2001 (i.e. $2 * (\text{region } 3) - 2 * (\text{region } 4) + ((\text{log average pay}) - 10)$). The time trends, i.e.$Y_{post}(0) - Y_{pre}(0)$, is generated by $0.1 * (\text{log average pay}) + 0.1 * (\text{region } 3) + 0.1 * (\text{years after treatment}) + (\text{region } 4) * (\text{years after treatment})^2 + (\text{log average pay})^{\frac{1}{2}} * (\text{log average population})$. The treatment effect is defined as $0.1 * (\text{log average population}) + 0.1 * (\text{log average population})^{\frac{1}{2}}$.

## E.2. Additional Results

*Table 4.* MSE (mean ± standard deviation) over 100 simulations following the conditional parallel trends condition. Each row represent a different meta-learner, and columns represent the different nuisance function classes.

|  | Basic | Lasso (CV) | Ridge (CV) | Random Forest | Best |
|---|---|---|---|---|---|
| Neural Net (CPTA OR) | 0.12 ± 0.02 | 0.12 ± 0.02 | 0.12 ± 0.02 | 0.38 ± 0.18 | 0.12 ± 0.02 |
| Neural Net (CPTA DR) | 0.1 ± 0.02 | 0.1 ± 0.03 | 0.1 ± 0.02 | 0.14 ± 0.04 | 0.1 ± 0.02 |
| Neural Net (Lagged OR) | 0.12 ± 0.02 | 0.14 ± 0.04 | 0.12 ± 0.02 | 1.27 ± 0.65 | 0.12 ± 0.02 |
| Neural Net (Lagged DR) | 0.1 ± 0.02 | 0.1 ± 0.03 | 0.1 ± 0.02 | 0.63 ± 0.4 | 0.1 ± 0.02 |
| XGBoost (OR) | 0.09 ± 0.02 | 0.09 ± 0.02 | 0.09 ± 0.02 | 0.31 ± 0.16 | 0.09 ± 0.02 |
| XGBoost (DR) | 0.04 ± 0.01 | 0.04 ± 0.01 | 0.04 ± 0.02 | 0.06 ± 0.03 | 0.04 ± 0.01 |
| XGBoost (Lagged OR) | 0.09 ± 0.02 | 0.11 ± 0.04 | 0.09 ± 0.02 | 1.15 ± 0.69 | 0.09 ± 0.02 |
| XGBoost (Lagged DR) | 0.04 ± 0.01 | 0.05 ± 0.03 | 0.04 ± 0.02 | 0.54 ± 0.45 | 0.04 ± 0.01 |
| Linear (OR) | 0.26 ± 0.07 | 0.26 ± 0.07 | 0.26 ± 0.07 | 0.51 ± 0.18 | 0.26 ± 0.07 |
| Linear (DR) | 0.26 ± 0.07 | 0.26 ± 0.07 | 0.26 ± 0.07 | 0.26 ± 0.07 | 0.26 ± 0.07 |
| Linear (Lagged OR) | 0.26 ± 0.07 | 0.28 ± 0.08 | 0.26 ± 0.07 | 1.18 ± 0.56 | 0.26 ± 0.07 |
| Linear (Lagged DR) | 0.26 ± 0.07 | 0.26 ± 0.07 | 0.26 ± 0.07 | 0.42 ± 0.19 | 0.26 ± 0.07 |

*Table 5.* MSE (mean ± standard deviation) Over 100 Simulations of Imbalanced Dataset. Each row represent a different meta-learner, and columns represent the different nuisance function classes.

|  | No Controls | Linear Regression | Lasso (CV) | Ridge (CV) | Random Forest | Best |
|---|---|---|---|---|---|---|
| Neural Net (OR) | 1.53 ± 0.74 | 0.22 ± 0.06 | 0.21 ± 0.06 | 0.21 ± 0.06 | 0.4 ± 0.15 | 0.21 ± 0.05 |
| Neural Net (DR) | 0.52 ± 0.31 | 0.18 ± 0.07 | 0.18 ± 0.05 | 0.18 ± 0.05 | 0.24 ± 0.07 | 0.18 ± 0.05 |
| Neural Net (CATE OR) | 0.66 ± 0.27 | 0.27 ± 0.08 | 0.27 ± 0.08 | 0.27 ± 0.08 | 0.51 ± 0.16 | 0.27 ± 0.08 |
| Neural Net (CATE DR) | 0.53 ± 0.22 | 0.22 ± 0.07 | 0.22 ± 0.07 | 0.21 ± 0.07 | 0.33 ± 0.11 | 0.21 ± 0.07 |
| XGBoost (OR) | 1.22 ± 0.58 | 0.21 ± 0.06 | 0.21 ± 0.06 | 0.21 ± 0.06 | 0.34 ± 0.11 | 0.21 ± 0.06 |
| XGBoost (DR) | 0.4 ± 0.14 | 0.12 ± 0.03 | 0.12 ± 0.03 | 0.12 ± 0.03 | 0.18 ± 0.06 | 0.12 ± 0.03 |
| XGBoost (CATE OR) | 0.66 ± 0.27 | 0.27 ± 0.08 | 0.27 ± 0.08 | 0.27 ± 0.08 | 0.51 ± 0.16 | 0.27 ± 0.08 |
| XGBoost (CATE DR) | 0.49 ± 0.22 | 0.15 ± 0.05 | 0.15 ± 0.04 | 0.15 ± 0.05 | 0.34 ± 0.13 | 0.15 ± 0.04 |

*Table 6.* MSE (mean ± standard deviation) over 100 simulations following the conditional parallel trends condition, with 100 covariates.

|  | Linear Regression | Lasso (CV) | Ridge (CV) | Random Forest | Best |
|---|---|---|---|---|---|
| Neural Net OR | 0.21 ± 0.05 | 0.21 ± 0.05 | 0.2 ± 0.06 | 1.27 ± 0.69 | 0.21 ± 0.06 |
| Neural Net DR | 0.18 ± 0.05 | 0.18 ± 0.06 | 0.18 ± 0.06 | 0.64 ± 0.38 | 0.18 ± 0.06 |
| Neural Net CATE OR | 0.28 ± 0.08 | 0.28 ± 0.08 | 0.28 ± 0.08 | 0.65 ± 0.25 | 0.28 ± 0.08 |
| Neural Net CATE DR | 0.3 ± 0.1 | 0.29 ± 0.09 | 0.29 ± 0.1 | 1.08 ± 0.71 | 0.2 ± 0.06 |
| Linear OR | 0.27 ± 0.08 | 0.27 ± 0.08 | 0.27 ± 0.08 | 1.3 ± 0.63 | 0.27 ± 0.08 |
| Linear DR | 0.27 ± 0.08 | 0.27 ± 0.08 | 0.27 ± 0.08 | 0.44 ± 0.13 | 0.27 ± 0.08 |
| Linear CATE OR | 0.28 ± 0.08 | 0.27 ± 0.08 | 0.28 ± 0.08 | 0.65 ± 0.25 | 0.27 ± 0.08 |
| Linear CATE DR | 0.29 ± 0.08 | 0.29 ± 0.08 | 0.29 ± 0.08 | 0.82 ± 0.33 | 0.28 ± 0.08 |
| XGBoost OR | 0.21 ± 0.06 | 0.2 ± 0.05 | 0.21 ± 0.05 | 0.96 ± 0.45 | 0.21 ± 0.05 |
| XGBoost DR | 0.13 ± 0.04 | 0.12 ± 0.03 | 0.12 ± 0.03 | 0.61 ± 0.25 | 0.12 ± 0.03 |
| XGBoost CATE OR | 0.28 ± 0.08 | 0.27 ± 0.08 | 0.28 ± 0.08 | 0.65 ± 0.25 | 0.27 ± 0.08 |
| XGBoost CATE DR | 0.26 ± 0.09 | 0.24 ± 0.08 | 0.25 ± 0.09 | 1.18 ± 0.85 | 0.15 ± 0.05 |

*Table 7.* MSE (mean ± standard deviation) over 100 simulations that does not satisfy the conditional parallel trends assumption. Each row represent a different meta-learner, and columns represent the different nuisance function classes.

| | Linear Regression | Lasso (CV) | Ridge (CV) | Random Forest | Best |
|---|---|---|---|---|---|
| Neural Net (CPTA OR) | 76.93 ± 135.25 | 76.34 ± 129.71 | 74.87 ± 127.94 | 35.49 ± 91.98 | 36.85 ± 87.85 |
| Neural Net (CPTA DR) | 17.07 ± 74.16 | 15.54 ± 54.25 | 20.41 ± 86.35 | 17.24 ± 63.37 | 18.38 ± 65.13 |
| Neural Net (Lagged OR) | 70.31 ± 98.19 | 70.07 ± 93.93 | 69.98 ± 100.44 | 26.21 ± 39.93 | 24.81 ± 32.88 |
| Neural Net (Lagged DR) | 4.37 ± 4.66 | 4.93 ± 5.65 | 4.94 ± 5.89 | 4.99 ± 14.46 | 5.09 ± 10.25 |
| XGBoost (CPTA OR) | 65.67 ± 122.65 | 63.5 ± 127.89 | 63.82 ± 113.59 | 29.39 ± 69.84 | 30.15 ± 85.27 |
| XGBoost (CPTA DR) | 20.49 ± 58.55 | 22.99 ± 81.52 | 23.31 ± 74.74 | 26.9 ± 128.52 | 31.11 ± 149.05 |
| XGBoost (Lagged OR) | 55.62 ± 82.27 | 56.87 ± 83.95 | 53.95 ± 77.19 | 21.29 ± 32.85 | 22.39 ± 38.38 |
| XGBoost (Lagged DR) | 9.88 ± 14.34 | 9.34 ± 13.13 | 10.33 ± 21.81 | 10.76 ± 40.39 | 8.14 ± 13.38 |
| Linear (CPTA OR) | 18.41 ± 62.54 | 18.0 ± 61.84 | 18.41 ± 62.68 | 17.61 ± 65.51 | 17.61 ± 65.51 |
| Linear (CPTA DR) | 14.56 ± 57.69 | 14.7 ± 58.43 | 14.56 ± 57.7 | 15.84 ± 64.12 | 15.84 ± 64.12 |
| Linear (Lagged OR) | 12.08 ± 28.93 | 11.64 ± 27.55 | 12.07 ± 28.9 | 9.78 ± 21.18 | 9.78 ± 21.18 |
| Linear (Lagged DR) | 4.78 ± 5.81 | 4.85 ± 5.97 | 4.78 ± 5.81 | 3.99 ± 7.18 | 3.99 ± 7.18 |

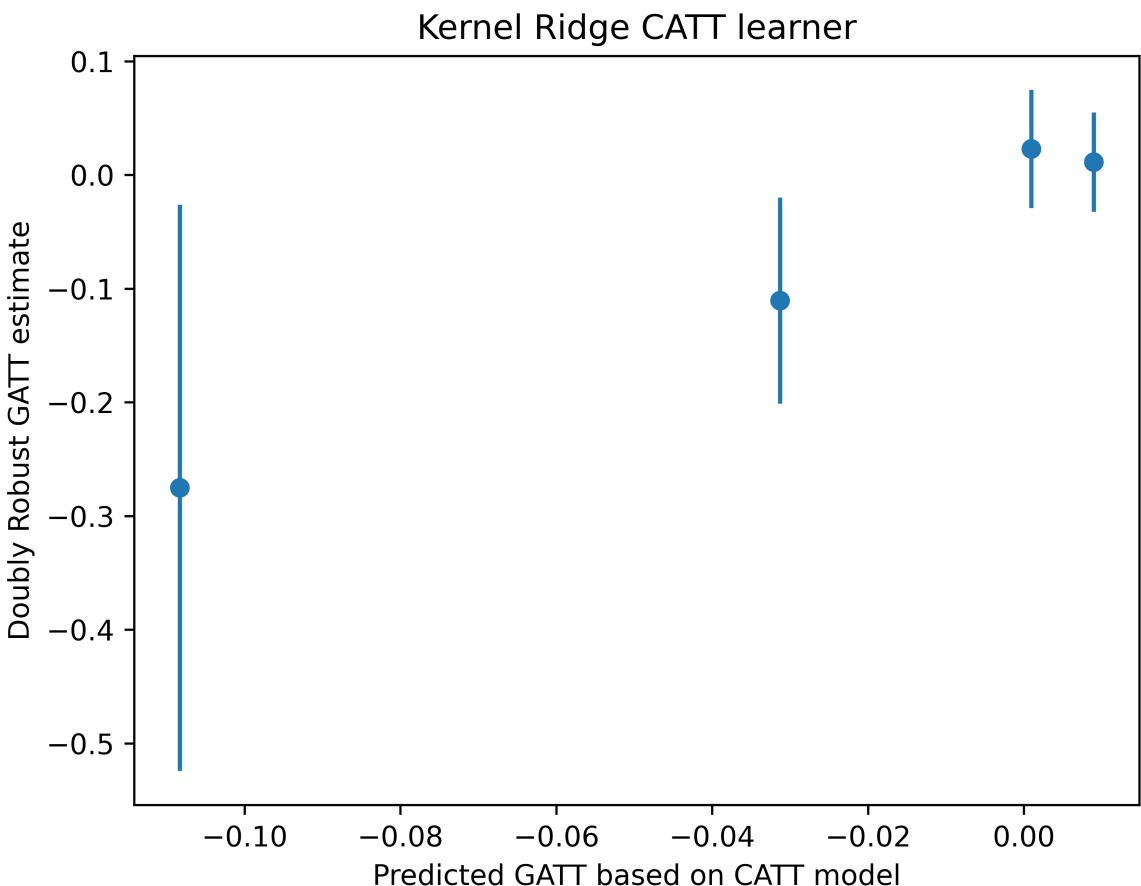

*Figure 3.* Calibration plot for CATT w.r.t log county population for the XGBoost doubly robust learner.

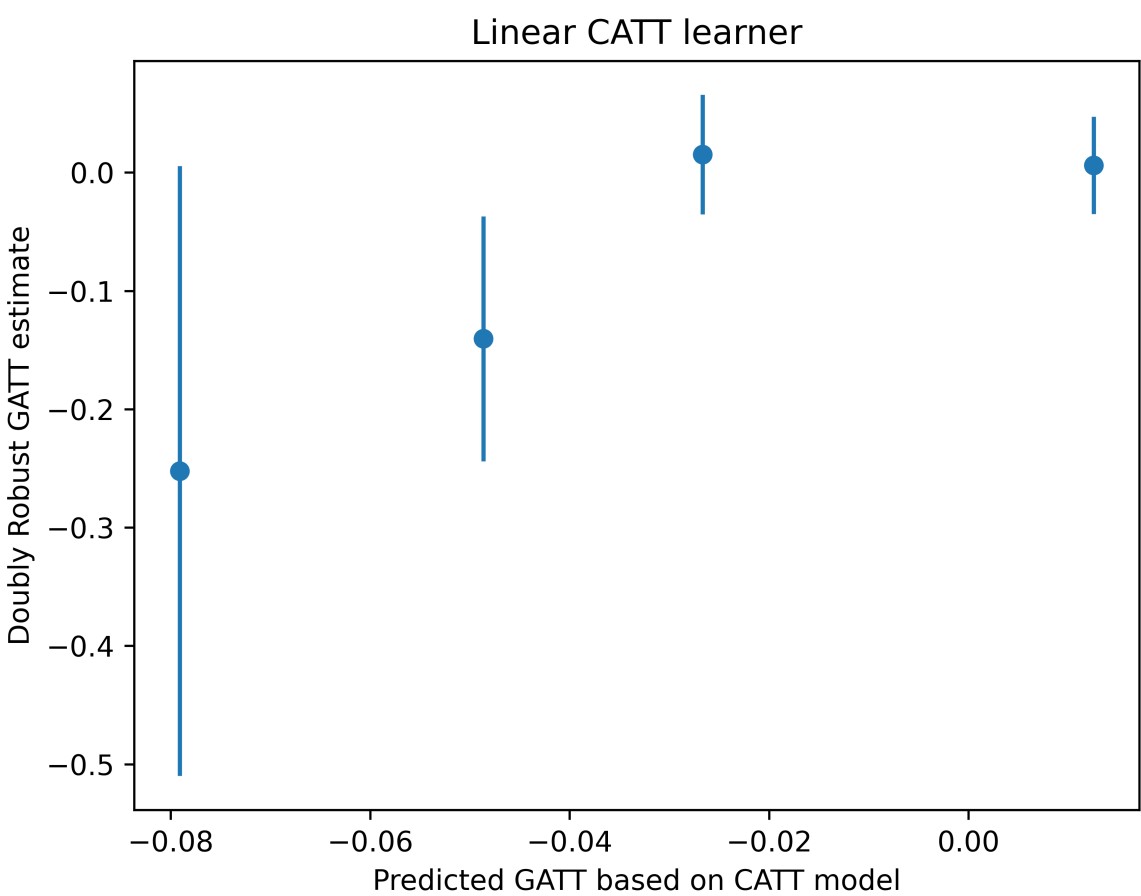

*Figure 4.* Calibration plot for CATT w.r.t log county population for the linear doubly robust learner.

