# OpenReview forum: "A Meta-learner for Heterogeneous Effects in Difference-in-Differences"
_ICML.cc/2025/Conference — ICML 2025 poster_

### Official Review · Reviewer_r9fP · 2025-03-06

**Overall Recommendation:** 4

**Summary:**

The paper introduces novel orthogonal meta-learners for estimating the conditional average treatment effect of the treated (CATT) in a difference-in differences (DID) setting. The main contributions include the proposed orthogonal loss, theoretical guarantees in form of doubly robust rates, and extensions to generalized settings. The proposed learner is evaluated using synthetic and real-world data.

**Claims And Evidence:**

The claims (orthogonality, double robustness) are clearly formulated and backed up with both theoretical and empirical evidence.

Only one question: is the identifiability result from Proposition 2.3. novel? If not, a clear reference would be needed.

**Essential References Not Discussed:**

See above.

**Experimental Designs Or Analyses:**

Results are mostly on synthetic data as common in causal inference (evaluation on real-world data is often impossible). The case-study on real-world data is in my opinion not entirely necessary (could be moved to the appendix).

**Methods And Evaluation Criteria:**

Evaluation criteria follow standard practices in the causal inference literature.

**Other Comments Or Suggestions:**

It would be nice if the authors could add a discussion section to the camera-ready version of the paper, discussion limitations and future work (e.g., as above).

**Other Strengths And Weaknesses:**

Strengths:
- Versatile framework that allows the use of arbitrary machine learning models for causal inference DID settings with nice theoretical guarantees

Weaknesses:
- While the meta-learner is novel, the derivation follows an established framework that has been leveraged for many other causal estimands already. New ideas remain relatively limited.
- As is common for doubly robust learners, the proposed learner may become unstable in low-overlap settings. It might be interesting to extend the methodology to overlap-weighted orthogonal learners (e.g., R-learner for DID).

In sum, a nice paper leveraging an existing framework for a new causal estimand. The result is a solid methodology that fills a gap in the literature.

**Questions For Authors:**

No further comments

**Relation To Broader Scientific Literature:**

Key literature seems to be cited (even though I am not an expert when it comes to DID).

**Theoretical Claims:**

I did not verify the correctness of the theoretical results in detail. However, I am quite familiar with similar work on orthogonal learning and the results look reasonable to me.

---

> ### Author Rebuttal · Authors · 2025-03-29
>
> Thank you for the helpful reviews! The result of Proposition 2.3 follows as a simple extension of the ATT identification strategy under conditional parallel trends. The reviewer is right that we should not claim this as a result of this work. We will reference for instance, in the preamble of the theorem the recent survey of [1], where a variant of this formula (when $X=W$) also appears (e.g. Equation (11) in [1]). The extra projection step on $W$ is a simple adaptation of that formula.  Regarding to the comment that there are limited new ideas, while we acknowledge that our methods stems from the idea of debiasing the nuisance functions, we want to highlight that our method provides a framework that allows one to debias nuisance functions that is learned on a different distribution (i.e. under covariate shift) from the target distribution for the final estimand, and this allows us to generalized the framework to many other estimands as described in Section 4.
>
> As pointed out by many reviewers, we were not able to include a detailed discussion section in the initial submission due to length constraints. In the camera-ready version, we will provide more discussions on the limitations of the proposed method including the unstable performance in low-overlap regimes, which is common for doubly robust learners. As you have mentioned, it is an interesting future direction to looking into overlap-weighted orthogonal learners. If one is interested in estimating overlap weighted projections of the true CATT, then we suspect that a simple adaptation of our loss, where one weights each sample by an appropriate weighting function of $W$, will cancel out the large variance of the density ratio, while maintaining Neyman orthogonality of the loss and will lead to a weighted MSE projection guarantee, weighted by the function of W that was used to reweight the loss at the sample level.
>
> Finally, we really appreciate the time you put into reviewing our paper, and the constructive feedback that will help us improve our work!
>
> **References**
> - [1] Roth, Jonathan, et al. "What’s trending in difference-in-differences? A synthesis of the recent econometrics literature." Journal of Econometrics 235.2 (2023): 2218-2244.

---

### Official Review · Reviewer_QLbJ · 2025-03-12

**Overall Recommendation:** 4

**Summary:**

The paper proposes a doubly robust metalearner for the Conditional Average Treatment Effect on the Treated for panel data. The method is targeted for binary
treatments and builds upon the parallel trends assumptions common in Difference-in-Differences settings.
Furthermore, the paper proposes
a robust meta-learning framework for the estimation of general conditional functionals under covariate shift.

**Claims And Evidence:**

Yes

**Essential References Not Discussed:**

/

**Experimental Designs Or Analyses:**

Yes

**Methods And Evaluation Criteria:**

Yes. However, experiments are only conducted on the setting with a specific treatment timestep, i.e., T=2. Empirical evaluation of the method for greater
time horizons would be of interest.

**Other Comments Or Suggestions:**

- Line 121: Typo "problem of estimating"
- Theorem 3.6: R should be defined
- Theorem 4.7: typo in title
- Please format the paper in a revised version: The running title is still the one from the template; Figures and tables are not were they are supposed to be
- The supplementary material has many typos

**Other Strengths And Weaknesses:**

**Strengths:**
- The idea is novel and relevant.
- The paper is well written and, in most parts, easy to follow for readers with sufficient background.

**Weaknesses:**
- The method assumes that all covariates are time-invariant. However, this is barely the case in real-world settings.
- The paper does not discuss the proposed method in terms of limitations or future work
- The paper does not provide a general literature review, making it difficult for readers not very familiar with orthogonal (CATE) meta-learners to assess
the differences to existing works and the novelty of the proposed method

**Questions For Authors:**

- Definition 3.1: What is the difference to Neyman-orthogonality and why is conditional Neyman-orthogonality necessary here? In my opinion, the text in the
definition describes normal NO, whereas the formula describes conditional NO.
- What is the benefit of the Lagged Dependent Outcome Alternate Assumption? Why is this introduced here?

**Relation To Broader Scientific Literature:**

The paper is related to a recent literature stream on Neyman-orthogonal learning for various causal effects (e.g., the popular CATE DR- or R-learner). It extends
the idea of constructing such an orthogonal meta-learner to estimate the conditional average treatment effect on the treated on panel data.

**Theoretical Claims:**

The proofs of the claims in the main paper.

---

> ### Author Rebuttal · Authors · 2025-03-29
>
> Thank you for the helpful reviews! We apologize for any typos and will definitely fix them for the camera-ready version. As pointed out by many reviewers, we were not able to include a detailed discussion and literature review sections in the initial submission due to length constraints. In the camera-ready version, we will expand on the related literature on CATE meta-learners to provide more background, and also provide more discussions on the limitations including the assumption that the covariates are time-invariant. First, we acknowledge that this is often not the case in real world applications. In such cases, one solution is to only use the pre-treatment covariates as in [1]. Alternatively, if there are other time varying variables that are also believed to be important for the time trends, one solution is to concatenate the covariates at the two different time points, and include all of them in the conditioning set for the conditional parallel trends conditions. But one must be careful to remove any covariates that are downstream from the treatment (i.e. descendents of the treatment in the context of a causal graph, these can be potential mediators) for the validity of the causal interpretation of the CATT.
>
> Regarding the Lagged Dependent Outcome assumption - this is also a common assumption that is widely used in economics when analysing panel data [2]. Depending on the context, this assumption might be more plausible than the parallel trends assumption, e.g. salary increase might be strongly correlated with the baseline salary level. While the parallel trends assumption might be seen as a special case of this assumption, many think of them as different assumptions, and the difference in the identification will result in different doubly robust estimation procedures when learning the conditional ATT on the full set of covariates $X$. In the case where the covariates of interest are a strict subset of the full set $X$, they can be estimated using the same framework. Hence, we include this setting in the appendix for comprehensiveness.
>
> In Section 5 of the paper, we discussed how to extend the learner to the multiple-time period settings. One solution is to learn a separate model for each lag period as in [1]. In this case, looking at more time periods is essentially the same task as in the two time period setting. An alternative would be to treat the lag period as an additional covariate to include in the conditioning set for the conditional parallel trends assumption. Practically this is what people can do. Qualitatively the insights from our current work should extend to this setting, even though the data are no longer i.i.d.; we believe that this also allows one to study the heterogeneity of the ATT with respect to the lag-periods. In this framework, the lag period may be seen as any other covariate used in the model. For this reason, we believe that results from simulation experiments in the two time-period setting should also generalize to the multiple time period setting. Note that our main theorems 3.6 and 4.7 do not really depend on the i.i.d. nature of the data. They are meta-theorems that state that as long as we have an ML algorithm that achieves a plug-in excess risk bound, then this translates to a mean-squared-error bound for the target causal estimand. Whether data are i.i.d. or not would be important when proving plug-in excess risk bounds, which would be easier for i.i.d., while for time-series or panel datasets, one would need to invoke martingale analysis to prove such excess risk bounds. Irrespective, our robustness conclusion will remain valid and is versatile, allowing for an arbitrary argument for the plug-in excess risk.
>
> On the clarification of conditional Neyman orthogonality we point out our response to Reviewer q3LX, who raised the same question.
>
> Finally, we really appreciate the time you put into reviewing our paper, and the constructive feedback that will help us improve our work!
>
> **References**
> - [1] Callaway, B., & Sant’Anna, P. H. (2021). Difference-in-differences with multiple time periods. Journal of econometrics, 225(2), 200-230.
> - [2] Angrist, J. D., & Pischke, J. S. (2009). Mostly harmless econometrics: An empiricist's companion. Princeton university press.
> - [3] Foster, Dylan J., and Vasilis Syrgkanis. "Orthogonal statistical learning." The Annals of Statistics 51.3 (2023): 879-908.

---

> > ### Comment · Reviewer_QLbJ · 2025-04-02
> >
> > Thank you for the clarification! I will raise my score to 4.

---

### Official Review · Reviewer_q3LX · 2025-03-14

**Overall Recommendation:** 4

**Summary:**

The paper develops a Neyman-orthogonal meta learner for estimating a conditional average treatment effect on the treated (CATT) in the framework of difference-in-differences (DiD). This framework casts the problem of CATT estimation as a convex risk minimisation that involves auxiliary (nuisance) models. The authors leveraged Neyman-orthogonality to further make the framework robust to the misspecification of the auxiliary models. Additionally, the authors provided several extensions of the method to (1) general functionals under covariate shift and (2) an instrumental DiD setting with non-compliance. At the end, the authors empirically demonstrated the superiority of their method.

**Claims And Evidence:**

The claims in the paper seem valid and are well supported. The authors provided the proofs for all the theoretical statements in the paper. Also, they provided the empirical evidence that the proposed approach is superior to existing methods.

**Essential References Not Discussed:**

It is not clear, why the conditional Neyman-orthogonality is required for estimating CATT and the regular (marginal) Neyman-orthogonality wouldn’t suffice (it seems to me that the conditional Neyman-orthogonality is a strictly stronger condition). Many existing learners like DR-/R-learners for CATE only rely on the marginal Neyman-orthogonality [1, 2], where in [2], any subset $W \subseteq X$ can be considered as the second-stage model input. Hence, I strongly encourage the authors to provide the discussion on why the conditional Neyman-orthogonality (and conditional Riesz Representer) are essential for this work.

References:
- [1] Foster, Dylan J., and Vasilis Syrgkanis. "Orthogonal statistical learning." The Annals of Statistics 51.3 (2023): 879-908.
- [2] Morzywolek, Pawel, Johan Decruyenaere, and Stijn Vansteelandt. "On a general class of orthogonal learners for the estimation of heterogeneous treatment effects." arXiv preprint arXiv:2303.12687 (2023).

**Experimental Designs Or Analyses:**

See Methods And Evaluation Criteria.

**Methods And Evaluation Criteria:**

The contribution of the paper is rather theoretical. Yet, as a causal ML paper, the work follows a well-established practice of (semi-)synthetic evaluation.

**Other Comments Or Suggestions:**

I found some notational inconsistencies (e.g., the lower indices of $D$ in all the sections). I encourage the authors to carefully check the notation and fix it during the rebuttal.

**Other Strengths And Weaknesses:**

Strength:
- I really like the scope of the paper and its extensions to the covariate shifts and the instrumental DiD settings.

Weaknesses (can be fixed during the rebuttal):
- I found the structure of the paper slightly confusing. The authors often switch back and forth between (i) assumptions, (ii) identification formulas, and (iii) estimation approaches (e.g. in Sec. 3 and 5). I would suggest having a clear cut between all three and present the corresponding sections in the sequential manner.

**Questions For Authors:**

- Do we need a new term for the nuisance functions: auxiliary models?

**Relation To Broader Scientific Literature:**

The proposed approach fills the gap of meta learners in the DiD setting. The authors also extended it further to the covariate shift problem and the instrumental DiD.

**Theoretical Claims:**

I did not carefully check the proofs, but the overall approach follows the Neyman-orthogonality theory from [1].

References:
- [1] Foster, Dylan J., and Vasilis Syrgkanis. "Orthogonal statistical learning." The Annals of Statistics 51.3 (2023): 879-908.

---

> ### Author Rebuttal · Authors · 2025-03-29
>
> Thank you for the helpful reviews! We will fix the inconsistent notations and restructure the paper for the camera-ready version. We introduced “auxiliary models” in the abstract to also accommodate for audiences that are not very familiar with the field. That being said, given the target audience, we might change it to “nuisance functions” for clarity.
>
> On the clarification of conditional Neyman orthogonality: in [1,2], neyman-orthogonality is referenced as a property of the loss, and is defined by:
>
> $$D_g D_{\theta} \mathcal{L} (\theta^*, g_0)[\theta - \theta^*, g - g_0] = 0 \quad \forall \quad \theta \in \Theta, \quad g \in \mathcal{G}$$
>
> In Definition 3.2 of our paper, Neyman orthogonality is introduced as the property of the moment (see also [3]). Since it is a conditional moment restriction, it is natural that the directional derivatives vanish even conditioning on $W$. A conditionally Neyman orthogonal moment implies a Neyman orthogonal loss, if the gradient of the loss is associated with the moment. Thus conditional Neyman orthogonality of the moment suffices for a Neyman orthogonal loss, but as the reviewer suggests it is not necessary. However, if the target model space Theta is completely unstructured, then the two become equivalent.
>
> $$D_g D_{\theta} \mathcal{L} (\theta^*, g_0)[\theta - \theta^*, g - g_0]
> = 2D_g \mathbb{E} [m(\theta, g_0) (\theta - \theta^*)] [g - g_0]
> = 2D_g \mathbb{E}\left[\mathbb{E} [m(\theta, g_0)|W] (\theta - \theta^*)\right] [g - g_0]$$
> where $m(\theta, g) = \widehat{Y}(g) - D\theta(W)$ is the score function for the conditional moment restriction that we proposed.
>
> Since this has to hold for an arbitrary $\theta$, we have $D_g D_{\theta} \mathcal{L} (\theta^*, g_0)[\theta - \theta^*, g - g_0] =0$ if and only if $D_g \mathbb{E}[m(\theta, g_0)|W] [g - g_0]=0 $ for all $g$ with measure 1 (since we can take $\theta=\theta^* + \delta$, where $\delta$ is the Dirac-delta function at a particular $W$). Similarly, since we are working with a conditional moment restriction (since the estimand is the CATT instead of the ATT), we also need the conditional Riesz Representer.
>
> Constructing Neyman orthogonal losses by first constructing a conditional Neyman orthogonal moment and finding the loss whose gradient is associated with this moment is a typical way of constructing an orthogonal loss. The meta-learners that the reviewer references can also be viewed through this lens. They correspond to Neyman orthogonal losses, because their gradient is associated with a moment restriction that satisfies conditional Neyman orthogonality (see e.g. the [4] for this fact, where also conditional Neyman orthogonality is used for both DR and R Learner style estimators of CATEs via forests). Moreover, this sufficient condition is satisfied for all the applications that our work is motivated by (e.g. DiD CATTs and all the applications of the general framework of conditional linear functionals with covariate shift).
>
> Finally, we really appreciate the time you put into reviewing our paper, and the constructive feedback that will help us improve our work!
>
> **References**
> - [1] Foster, Dylan J., and Vasilis Syrgkanis. "Orthogonal statistical learning." The Annals of Statistics 51.3 (2023): 879-908.
> - [2] Morzywolek, Pawel, Johan Decruyenaere, and Stijn Vansteelandt. "On a general class of orthogonal learners for the estimation of heterogeneous treatment effects." arXiv preprint arXiv:2303.12687 (2023).
> - [3] Chernozhukov, V., Chetverikov, D., Demirer, M., Duflo, E., Hansen, C., Newey, W., & Robins, J. (2018). Double/debiased machine learning for treatment and structural parameters.
> - [4] Oprescu, Miruna, Vasilis Syrgkanis, and Zhiwei Steven Wu. "Orthogonal random forest for causal inference." International Conference on Machine Learning. PMLR, 2019.

---

### Decision · Program_Chairs · 2025-05-01

**Decision:**

Accept (poster)

**Comment:**

The paper introduces a meta-learner for estimating heterogeneous effects in difference-in-differences designs.

I agree with the reviewers that additional discussions on the necessity of conditional Neyman orthogonality would improve the paper (e.g., including a condensed version of your reply to q3LX's review would be very helpful). Similarly, a short discussion of the low-overlap setting (raised by r9fP) would be useful.

I also tend to agree with Reviewer r9fP's assessment of the novelty ("the derivation follows an established framework that has been leveraged for many other causal estimands already. New ideas remain relatively limited"). I believe that careful consideration of the multiple time point design and more complex designs (e.g., staggered difference-in-differences) would enhance the overall significance/impact of the work. Your response to r9fP makes me hopeful that such extensions are possible and I would encourage you to explore them a bit further.